# Multi-layer Cloud Conditions in Trade Wind Shallow Cumulus – Confronting two ICON Model Derivatives with Airborne Observations

Marek Jacob[1], Pavlos Kollias[1,2], Felix Ament[3], Vera Schemann[1], and Susanne Crewell[1]

[1]Institute for Geophysics and Meteorology, University of Cologne, Albertus-Magnus-Platz, 50923 Cologne, Germany
[2]School of Marine and Atmospheric Sciences, Stony Brook University, Stony Brook, NY 11794-5000, USA
[3]Meteorological Institute, Universität Hamburg, Bundesstrasse 55, 20146 Hamburg, Germany

**Correspondence:** Marek Jacob (marek.jacob@uni-koeln.de)

**Abstract.** Airborne remote sensing observations over the tropical Atlantic Ocean upstream of Barbados are used to characterize trade wind shallow cumulus clouds and to benchmark two cloud-resolving ICON (ICOsahedral Nonhydrostatic) model simulations at kilo- and hectometer scales. The clouds were observed by an airborne nadir pointing backscatter lidar, a cloud radar, and a microwave radiometer in the tropical dry winter season during daytime. For the model benchmark, forward operators convert the model output into the observational space for considering instrument specific cloud detection thresholds. The forward simulations reveal the different detection limits of the lidar and radar observations, i.e., most clouds with cloud liquid water content greater than $10^{-7}\,\mathrm{kg\,kg^{-1}}$ are detectable by the lidar, whereas the radar is primarily sensitive to the "rain"-category hydrometeors in the models and can detect even low amounts of rain.

The observations reveal two prominent modes of cumulus cloud top heights separating the clouds into two layers. The lower mode relates to boundary layer convection with tops closely above the lifting condensation level, which is at about $700\,\mathrm{m}$ above sea level. The upper mode is driven by shallow moist convection, also contains shallow stratiform outflow anvils, and is closely related to the trade inversion at about $2.3\,\mathrm{km}$ above sea level. The two cumulus modes are sensed differently by the lidar and the radar observations and under different liquid water path (LWP) conditions. The storm-resolving model (SRM) at kilometer scale reproduces the cloud modes barely and shows most cloud tops being slightly above the observed lower mode. The large-eddy model (LEM) at hectometer scale reproduces better the observed cloudiness distribution with a clear bimodal separation. We hypothesize that slight differences in the autoconversion parametrizations could have caused the different cloud development in the models. Neither model seems to account for in-cloud drizzle particles that do not precipitate down to the surface but generate a stronger radar signal even in scenes with low LWP. Our findings suggest that even if the SRM is a step forward for better cloud representation in climate research, the LEM can better reproduce the observed shallow cumulus convection and should therefore in principle represent cloud radiative effects and water cycle better.

# 1 Introduction

The representation of low-level oceanic clouds contributes largely to differences between climate models in terms of equilibrium climate sensitivity (Bony and Dufresne, 2005; Schneider et al., 2017). Global atmospheric models with kilometer-scale resolution are considered the way forward in forecasting future climate scenarios (Satoh et al., 2019). The increased model resolution and better matching scales with measurements allow for a more direct observational assessment by comparing the present day representation in the models with atmospheric measurements and thus anchoring models to reality. Recently, Stevens et al. (2020) demonstrated the general advantage of high resolution simulations compared to typical climate models in terms of cloud representation using different versions of the ICOsahedral Non-hydrostatic model (ICON). The progress in such novel large-area high-resolution models and new capabilities of synergetic airborne measurements in the trades motivate the following guiding questions of this study. How do two cloud-resolving versions of the ICON model represent shallow cumuli in comparison to observations? What is an appropriate approach to assess the model clouds? How does the liquid water path (LWP) help interpreting differences between observed and simulated cloud structures?

Increased model resolution facilitates the model-observation comparisons. However, there are several other factors to be considered (Lamer et al., 2018). On the one hand, particle size distributions (PSDs) in models are typically represented by bulk and spectral microphysical schemes, or Lagrangian superparticles (e.g., Grabowski et al., 2019). Bulk microphysics schemes predict changes in condensate using one to three moments. These are usually the lower moments like particle number concentration and mixing ratio (Khain et al., 2015). On the other hand, radars and lidar, like those used in this study, observe different moments of the PSD. A backscatter lidar, for example, is primarily sensitive to the second moment, while a radar is sensitive to the sixth moment.

An objective definition of a cloud is required when comparing cloudiness in models with observations. If one asks different instrument operators to provide average "cloud coverage", one can get different answers, e.g., 19 to 46 %, for the very same scene as demonstrated by Stevens et al. (2019). This range is caused by different sensitivities due to different measurement principles and sampling methods by the remote sensing instruments involved and also affects global climatologies (e.g., Stubenrauch et al., 2013). To find a common definition, it is favorable to compare clouds in models and observations in terms of the same quantities (Bodas-Salcedo et al., 2011). Here, forward simulators can be used to simulate measurements as they would be recorded by a radar or lidar, based on the atmospheric state and assumptions in the model (Lamer et al., 2018).

The observations used in this study were recorded with the research aircraft HALO (High Altitude and LOng range; Krautstrunk and Giez, 2012) which was equipped as a flying remote sensing cloud observatory during the NARVAL-South experiment (Next generation Advanced Remote sensing for VALidation; Klepp et al., 2014) in December 2013. A reason to initiate the NARVAL expeditions was to extend satellite observations. This can test which cloud variables are sufficiently resolved from space and which characteristics benefit from higher spatial resolution in respect to shallow cumulus clouds. For example, the spaceborne Cloud-Aerosol Lidar with Orthogonal Polarization (CALIOP) has frequently been used to investigate marine low clouds. Leahy et al. (2012) observed two modes of low clouds in the tropical Pacific trade wind, and reveal that CALIOP misses small clouds ( < 1 km) and combines adjacent but separated clouds due to the CALIOP sampling rate. Leahy et al.

identified modes at about 800 and 2000 m above the sea surface in tropical Pacific trade wind cumulus (15° S, 155° W) using two years of data. The shallower mode is considered to be formed by small cumuli with insufficient buoyancy to grow while convective clouds and detrained elements from thick clouds suppressed under subsidence form the deeper mode. Genkova et al. (2007) compared trade wind cumuli cloud top heights from passive optical spaceborne instruments. They identified two cloud top modes at 650 and 1500 m above sea level in an area similar to this study (10°–20° N to 55°–65° W) from about 150 scenes between September 2004 and March 2005 using data from three different satellites. However, they also found vertical biases of 250 to 500 m due to different retrieval approaches and spatial resolutions.

In addition to spaceborne, also ground-based observations have been used to study the distribution of low-level cloud in the trades. Nuijens et al. (2014) analyzed cloud observations taken at the Barbados Cloud Observatory (BCO; Stevens et al., 2015) at the upstream eastern coast of Barbados at Deebles Point facing the Atlantic Ocean. For two years of ceilometer data of shallow clouds with tops below 4 km, they found that the shallow cloudiness is dominated by clouds near the lifting condensation level (LCL) with about two thirds of the shallow cloud coverage coming from clouds with bases below 1 km.

Since active instruments are advantageous for observing cloud vertical extents, the HALO instrumentation included an aerosol backscatter lidar as part of the WALES (WAter vapor Lidar Experiment in Space; Wirth et al., 2009) airborne demonstrator, and a cloud radar. The radar is one part of the HAMP (HALO microwave package; Mech et al., 2014) while a microwave radiometer is the other. The latter provides the vertically integrated liquid water content (LWP) (Jacob et al., 2019), which helps to approach the liquid water content which is a key quantity to describe clouds in models like the ICON. The direct observation of the liquid water content profile is difficult (Crewell et al., 2009), but the LWP can be used to estimate the water content when combined with estimates of cloud vertical extend by lidar and radar either in a simple average approach or more sophisticated as a profile (Frisch et al., 1998; Küchler et al., 2018). In addition, dropsondes were released regularly during the flights to probe the temperature and humidity profile. Compared to ground-based observations, the airborne remote sensing instruments, especially the microwave radiometer, have the advantage of not being harmed by precipitation or sea spray deposition on the instrument (Rose et al., 2005).

The observations are used to confront the simulations of winter season trade wind cumuli in the tropical west Atlantic Ocean. Such clouds are regularly subject in idealized large-eddy simulation (LES) studies (e.g., Siebesma et al., 2003; van Zanten et al., 2011; Bretherton and Blossey, 2017) due to their high relevance for the climate. As it is difficult for small domain LES models to generate realistic mesoscale cloud organization (Jeevanjee and Romps, 2013), we use simulations by Klocke et al. (2017) that were run on large domains (> 1500 x 900 km) with kilo- and hectometer horizontal grid spacings and were forced by numerical weather prediction output. Simulations with 1.25 km grid spacing were produced using the storm-resolving model (SRM) version of ICON, while simulations with 3 hectometer grid spacing were produced using ICON large-eddy model (LEM).

To assess vertically resolved cloudiness and shallow convection, we compare the vertical cloud boundaries. The exact location of the cloud boundaries is the major parameter determining the heating rate profile. Further, the cloud top height is an indicator of the convective activity and therefore allows assessing the model physics with observations indirectly. While cloud fraction at cloud base is rather robust among model assumptions, cloud fraction near the trade inversion varies strongly (Vogel

et al., 2020). Clouds near the inversion are often very thin (O et al., 2018) and therefore the problem of instrument sensitivity (e.g., Stubenrauch et al., 2013) can provide different answers about their exact vertical placement. The use of instrument simulators applied to model output allows the different sensitivities of the instruments to be taken into account, as explained in the following.

As the backscatter lidar becomes completely attenuated by the presence of hydrometeors in a cloud quickly, lidar measurements and their forward simulations are considered for a cloud top height estimate only. The radar, however, can penetrate through the cloud and precipitation layers and thus provides estimates of cloud or precipitation base heights in addition to cloud top heights. As shallow cumulus convection is not expected to trigger at the same time and place in a model and reality, a statistical approach is adopted here, in which the airborne observations are compared to their model counterpart for different LWP regimes. In the LWP space it is possible to study microphysical cloud processes like the transition from non-precipitating to precipitating clouds.

This paper is structured as follows: The observations and their sensitivities in Sect. 2 are followed by a brief description of the model setup and output in Sect. 3. Then, the forward simulations are presented in Sect. 4 taking into account the instrument characteristics and specifications of model outputs. Finally, the model outputs of ICON SRM and LEM are confronted with the airborne observations in Sect. 5 including the analysis in LWP space. A summary and conclusions are given in Sect. 6.

## 2 Observations

The airborne measurements were taken during the NARVAL-South (also referred to as NARVAL1) field experiment in the tropical Atlantic east of Barbados. The NARVAL remote sensing package (Stevens et al., 2019) recorded data during 8 research flights in the tropical domain south of 20° N from 10 to 20 December 2013. The flight tracks are depicted in Fig. 1. A total of about 22 000 km of HALO along track observations with about 91 thousand profiles were sampled at a frequency of 1 Hz from altitudes between 13 and 14.5 km. Further details of the experiment and flight planning are provided by Klepp et al. (2014) and Konow et al. (2019). In this study we use the backscatter lidar cloud top height, the radar reflectivity factor $Z$, liquid water path (LWP) retrieved from microwave radiometer, and the LCL estimated from dropsondes. The remote sensing lidar, radar, and microwave radiometer were installed in a near-nadir pointing direction under the fuselage of the aircraft.

The LWP retrieval from the microwave radiometer has a high accuracy, which is better than $20\,\mathrm{g\,m^{-2}}$ for LWP $< 100\,\mathrm{g\,m^{-2}}$ and relatively better than 20 and 10 % of the retrieved LWP for LWP greater than $100\,\mathrm{g\,m^{-2}}$ and $500\,\mathrm{g\,m^{-2}}$, respectively, as described by Jacob et al. (2019). The LWP is defined as the integral of all liquid in the column comprising cloud liquid and rain water. The LCL is derived from the dropsonde temperature and relative humidity (RH) measurements closest to the surface using the code by Romps (2017). The LCL measurement uncertainty is mostly affected by the RH measurement, such that an overestimation on the order of the calibration repeatability of 2 % RH (Vaisala, 2017) would result in an about 60 m lower LCL. The LCL from dropsonde releases is temporally interpolated to generate a continuous time series along the flight track. Fifty dropsondes were released in total in the study area with a median separation along flight track of about 515 km (quartiles:

384 and 658 km, see also Fig. 1). The following subsections describe the measurement principles of the radar and lidar and the respectively used thresholds for cloud detection.

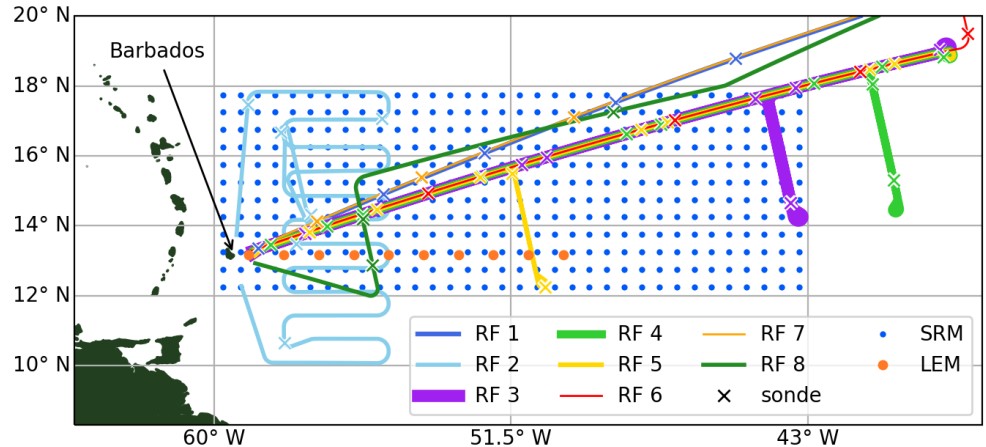

**Figure 1.** Map showing research flight (RF) tracks and the model columns, which are used in this study. The storm-resolving model (SRM, blue, original model grid spacing: 1.25 km) is thinned to a $0.5° \times 0.5°$ grid. From the large-eddy model (LEM, orange, original model grid spacing: 300 m), ten meteogram outputs are used.

## 2.1 Radar

The radar reflectivity factor – short "reflectivity" – $Z$ is measured by the HAMP radar at 35.5 GHz. In case of small spherical liquid droplets, the radar reflectivity is approximately proportional to the sixth moment of the PSD at a given range. This means that larger raindrops show a higher reflectivity than smaller cloud droplets given the same mass mixing ratio. The HAMP radar is calibrated following Ewald et al. (2019) and was operated at a vertical resolution of about 30 m with 1 Hz sampling. This sampling frequency corresponds to a surface footprint of about $136\,\text{m} \times 376\,\text{m}$ at a cruising speed of about $240\,\text{m\,s}^{-1}$.

The instruments minimal detectable signal (MDS) in dBZ decreases with range $r$ and is estimated by Ewald et al. (2019) as

$$\text{MDS}(r) = -39.8 + 20\log_{10}\left(\frac{r}{5\,\text{km}}\right). \tag{1}$$

According to this equation, the MDS in the shallow cumulus layer is about $-32$ dBZ when flying at 13 km. However, this does not include sensitivity reduction due to Doppler broadening caused by the aircraft motion (Mech et al., 2014). In this study, we use a sensitivity threshold of $-20$ dBZ by comparing HALO radar statistics with ground-based measurements taken at the BCO as outlined in the appendix A.

## 2.2 Lidar

The lidar system WALES supplements the HAMP radar with optical active remote sensing on HALO. WALES comprises a water vapor differential absorption lidar system (DIAL) at different wavelengths and a high spectral resolution lidar (HSRL)

which measures molecular and aerosol backscatter at 532 and 1064 nm. The scattering of an emitted laser pulse on a liquid hydrometeor mostly follows the principles of geometrical optics as the wavelength is much smaller than the particle. Therefore, the back-scattered energy is in first order approximation proportional to the hydrometeor cross section and thus to the second moment of the PSD (O'Connor et al., 2005). This means that a backscatter lidar is more sensitive to the number of small droplets compared to a radar. Besides hydrometeors, also other aerosol particles like dust scatter the lidar pulse back. However, the aerosol signal is much smaller than the hydrometeor signal. Therefore, we follow Gutleben et al. (2019) and use a threshold of backscatter ratio (BSR) > 20 in the 532 nm channel to differentiate cloudy scenes from clear-sky or dusty scenes. As cloud droplets attenuate the lidar signal strongly, the WALES lidar is used only to detect cloud top height using that threshold. This lidar top height is measured every second with a vertical accuracy of 15 m and the lidar footprint width at the surface is at about 22 m.

## 3   ICON-NARVAL model output

Two different versions of the ICON model were run to supplement the NARVAL experiment. The runs of the so-called storm-resolving model (SRM) and the large-eddy model (LEM) are described by Klocke et al. (2017) and Vial et al. (2019). The most important aspects relevant for this study of the SRM and LEM are summarized in this section.

### 3.1   ICON SRM

The SRM (Zängl et al., 2015) was run at 2.5 and 1.25 km horizontal grid spacing with a stretched vertical grid of 75 levels up to 30 km which has 12 and 22 levels below 800 m and 3 km, respectively. The domain spans the western tropical North Atlantic from 4° S to 18° N and from 64° W to 42° W (Stevens et al., 2019). The 1.25 km SRM is one-way nested into a coarser 2.5 km SRM which is initialized and nudged with lateral boundary data from the European Centre for Medium-Range Weather Forecasts (ECMWF). The SRM is run without a convection parameterization. The cloud and precipitation microphysics are represented by a one-moment microphysics scheme (Baldauf et al., 2011) that predicts the specific water contents of five different hydrometeor classes including liquid cloud water ($q_c$) and rain ($q_r$). In addition, a diagnostic sub-grid scheme adjusts the cloud fraction and cloud water input to the radiation code. In the following however, the cloud water predicted by the prognostic equations is considered primarily, but the potential influence of the diagnostic scheme is discussed in appendix B. 17 modeled days from 10 to 28 December 2013 are used and cover the whole NARVAL experiment. More details on the model setup can be found in Klocke et al. (2017).

The model output is archived hourly. The SRM was initialized at 00 UTC on each day. This study uses model output between 12:00 and 21:00 UTC (08:00 - 17:00 local time) granting the model 12 hr of spin-up and avoiding diurnal cycle influence. The data are spatially subsampled on a coarser $0.5° \times 0.5°$ grid to reduce the computational effort while still conserving the variety of atmospheric profiles. A compromise of domain overlap between all available model output and observations is achieved by limiting the SRM output to the area of 12 to 18° N and 60 to 43° W as marked in Fig. 1. If not specified differently, the

term "SRM" relates to the 1.25 km grid-spacing. The 2.5 km SRM output is only considered to discuss the influence of model resolution.

## 3.2 ICON LEM

The LEM (Dipankar et al., 2015; Heinze et al., 2017) with 300 m grid spacing was run in a multi-step nested setup, including a 600 m LEM nest, forced with the SRM . The LEM was initialized at 09 UTC of each simulation day using the outputs from the SRM. This means that the LEM also has a realistic, non-idealized initialization. The LEM vertical grid also reaches up to 30 km and has 150 levels with 14 and 37 of them below 800 m and 3 km, respectively. The LEM physics package differs from the SRM configuration. The LEM uses a Smagorinsky scheme for turbulence, but the most important difference for this study is that the microphysics are represented by the two-moment scheme of Seifert and Beheng (2001). This scheme predicts the hydrometeor number concentrations in addition to the specific water contents and thus provides $N_c$ and $N_r$ for liquid cloud water and rain, respectively.

In contrast to the SRM, the LEM was only run for the six days of research flights 2 to 6 and 8. However, the full hydrometeor state including rainwater and the number concentrations were only archived for four of the runs in the form of so-called "meteogram output". This means that hydrometeor profiles are available with high temporal resolution (every 36 s) but only at 12 model columns. Such meteogram output was saved for the days of research flights 4, 5, 6, and 8. The ten model columns east of Barbados are used for this study and are also marked in Fig. 1. The LEM output is also limited to the time between 12:00 and 21:00 UTC, after 3 hr of spin-up. The term "LEM" relates to the 300 m grid-spacing simulation if not specified differently. The 600 m LEM, however, is only considered for the assessment of horizontal resolution.

## 4 Radar and Lidar Forward Simulations

Forward simulators, also called forward operators, can simulate how the remote sensing instruments presented in Sect. 2 would perceive a scene provided by an atmospheric model. A forward simulator requires input like model variables and the knowledge about the microphysical assumptions employed in the atmospheric model. The basic variables are temperature, pressure, layer height, and humidity for each model level in a column for a 1D vertical forward simulation. The variables describing the hydrometeors depend on the microphysical scheme. Typically, these include mass mixing ratios (e.g., $q_c$ or $q_r$) of different hydrometeor classes. The forward simulator has to be configured such that the PSD used to simulate hydrometeor characteristics matches the PSD assumed in the atmospheric model accurately. This means that the same PSD shape and parameters have to be used in the simulator as assumed in the model. For models with advanced microphysical schemes, also the variables describing those aspects of the PSD are important input parameters for the forward simulation and need to be saved during the model run. In the case of the ICON LEM, the two-moment scheme by Seifert and Beheng (2001) uses the particle number concentrations as additional variables.

As this study focuses on the tropical shallow cumulus below freezing level, we confine the following description and analysis to precipitating and non-precipitating liquid hydrometeors, which are the raindrops and cloud droplets in the ICON microphysi-

cal schemes. Both ICON models assume modified Gamma distributions as PSDs. The number concentration $N(D)$ of spherical drops with diameter $D$ can be described as

$$N(D) = N_0\, D^\mu \exp(-\Lambda\, D^\gamma) \tag{2}$$

with the scale parameters $N_0$ and $\Lambda$ and the shape parameters $\mu$ and $\gamma$. These parameters are either fixed or derived from the input variables as described in Tab. 1.

**Table 1.** Configuration of modified Gamma distribution (Eq. 2) for liquid hydrometeors in ICON one and two-moment microphysical schemes.

| scheme | hydrometeor | $N_0$ | $\mu$ | $\Lambda$ | $\gamma$ | additional constrain |
|---|---|---|---|---|---|---|
| one moment (SRM) | cloud droplets | $f(q_c)$ | 8 | $f(q_c)$ | 3 | $N_c = 2 \times 10^8\,\mathrm{kg}^{-1}$ |
| one moment (SRM) | raindrops | $8 \times 10^6\,\mathrm{m}^{-4}$ | 0 | $f(q_r)$ | 1 | |
| two moments (LEM) | cloud droplets | $f(q_c, N_c)$ | 8 | $f(q_c, N_c)$ | 3 | |
| two moments (LEM) | raindrops | $f(q_r, N_r)$ | 2 | $f(q_r, N_r)$ | 1 | |

The lidar BSR is forward simulated using the lidar simulation capabilities of the Cloud Resolving Model Radar Simulator (CR-SIM; Oue et al., 2020). The code has been slightly modified such that the configuration for the two-moment ICON microphysics can be used for one-moment microphysics following the relations in Tab. 1. Though CR-SIM can also simulate radar reflectivity, the Passive and Active Microwave TRAnsfer package (PAMTRA; Mech et al., 2020) is used to forward simulate the radar as it offers a higher degree of flexibility.

The lidar forward simulations are used to detect the hydrometeors layer top and not for quantitative retrievals or estimates. Furthermore, as the airborne lidar is not affected by liquid collection on the telescope during raining conditions, there is no need to account for such effects, as there would be for a ground based lidar. As raindrops near cloud top are optically thin, if present in the models at all, we decided to simplify the forward simulation of the backscatter lidar and thus ignore their contributions. Therefore, the BSR is primarily a function of $q_c$ as shown in Figs. 2 a and b.

The approximated proportionality of the radar reflectivity to $D^6$ makes $Z$ especially sensitive to larger raindrops. Therefore, $q_r$ (and $N_r$) has to be considered in addition to $q_c$ (and $N_c$) when simulating the radar signal. The size difference between cloud droplets and raindrops produces a two-modal relation between the total liquid water concentration $q_t = q_c + q_r$ and $Z$ as it can be deduced from Figs. 2 c and d. The mode along a line of low $q_t$ corresponds to grid cells that predominantly feature rainwater. In this mode, even low amounts of liquid water in the rain category produce a reflectivity that can only be reached by cloud droplets with a three to four orders of magnitude higher cloud water content. Grid cells with such high $q_c$ and no $q_r$ align in a second mode parallel to the rain mode. A mixture of cloud and rain water accordingly results in an intermediate $Z$ which populates the space of $q_t > \approx 10^{-5}\,\mathrm{kg\,kg}^{-1}$ between the two main modes in Figs. 2 c and d. By setting the radar threshold to $-20\,\mathrm{dBZ}$, hardly any cloud-only grid cells in the lower right high-$q_t$ mode can be detected by the simulated radar. This means that in the forward-simulated dataset, all lidar-detectable hydrometeors are from the ICON cloud category while the radar-detectable hydrometeors have to contain at least a small amount of water from the ICON rain category.

The ICON LEM uses a two-moment scheme including $N_c$ and $N_r$. Therefore, the forward simulation broadens the relation between the water content and the forward-simulated signals (compare Figs. 2 a to b and c to d). The median of $N_c$ is $3.8 \times 10^8 \, \text{kg}^{-1}$ with an interquartile range between 2.0 an $6.8 \times 10^8 \, \text{kg}^{-1}$ for grid cells containing any cloud water. This means $N_c$ is mostly larger in the LEM than in the SRM (fixed to $N_c = 2 \times 10^8 \, \text{kg}^{-1}$, Tab. 1), and that cloud droplets are smaller for the same $q_c$ in the LEM. Thus, also the forward simulated signals are lower, such that slightly higher amounts of cloud water are required for a cloud to be detectable in the LEM than in the SRM. For a fixed water mixing ratio, a change by a factor of $\alpha$ in the number concentration ($N_c' = \alpha \, N_c$) will result to a change in the radar reflectivity in dBZ from $Z$ to $Z' = Z - 10 \log_{10} \alpha$. Thus, if we double the $N_c$ ($\alpha = 2$), the $Z$ will reduce by 3 dB. By the same token, if we change the hydrometeor diameter by a factor of $\alpha$ ($D' = \alpha \, D$), then the radar reflectivity will be $Z' = Z + 30 \log_{10} \alpha$. Different to cloud water, the radar reflectivity of rain is in many cases amplified in the two-moment scheme compared to the one-moment simulation, such that also some grid cells with lower $q_r$ are above the radar detection threshold. This indicates in general larger and fewer raindrops in the LEM than in the SRM for the same $q_r$ as also depicted in Fig. 3.

## 5  Model – observation comparison

Observations and forward simulations of the SRM and LEM runs are used to assess the vertical structures of the shallow clouds by focusing on the boundaries sensed by different instruments. In the following, shallow clouds are analyzed in terms cloud top heights estimated from lidar and radar measurements as well as the radar echo base height. All heights in the different scenes are set in relation to the theoretical cloud base of an adiabatic thermal-plume-driven boundary layer cloud by setting the height in relation to the LCL. First, a case study with example scenes from the observations and the LEM illustrates the approach. The case study is followed by the statistical analysis of the full datasets and the analysis stratified in the liquid water space to identify differences in microphysical processes.

To ease the following discussion, we define three layers in which the lidar and radar signals occur. Every signal below LCL is in the "precipitation" layer. Typically, only the radar base is in this layer. Clouds with their tops within $600 \, \text{m}$ above LCL are called "very shallow clouds" following the definitions by Vial et al. (2019). Vial et al. defined this mode in terms of an absolute top height below $1.3 \, \text{km}$ which corresponds to a similar height considering that the LCLs in the dropsonde, SRM, and LEM datasets in this study have typical heights of $720 \pm 135$, $763 \pm 144$, and $777 \pm 121 \, \text{m}$, respectively. Cumulus humilis is a typical representative of these very shallow clouds but in principle this class contains also small parts of deeper but slanted clouds. More active clouds can grow deeper than these very shallow clouds until they encounter the trade inversion and are forced to form a lateral outflow which is often perceivable as a stratiform layer. Stratiform remnants of such shallow convection can last for hours and thus much longer than the original convective core (Wood et al., 2018). We summarize all cloud signals above LCL + $600 \, \text{m}$ as "stratiform" mode, acknowledging also contributions from active cores. To limit the analysis to shallow clouds, an upper limit is set to $4 \, \text{km}$ above sea surface.

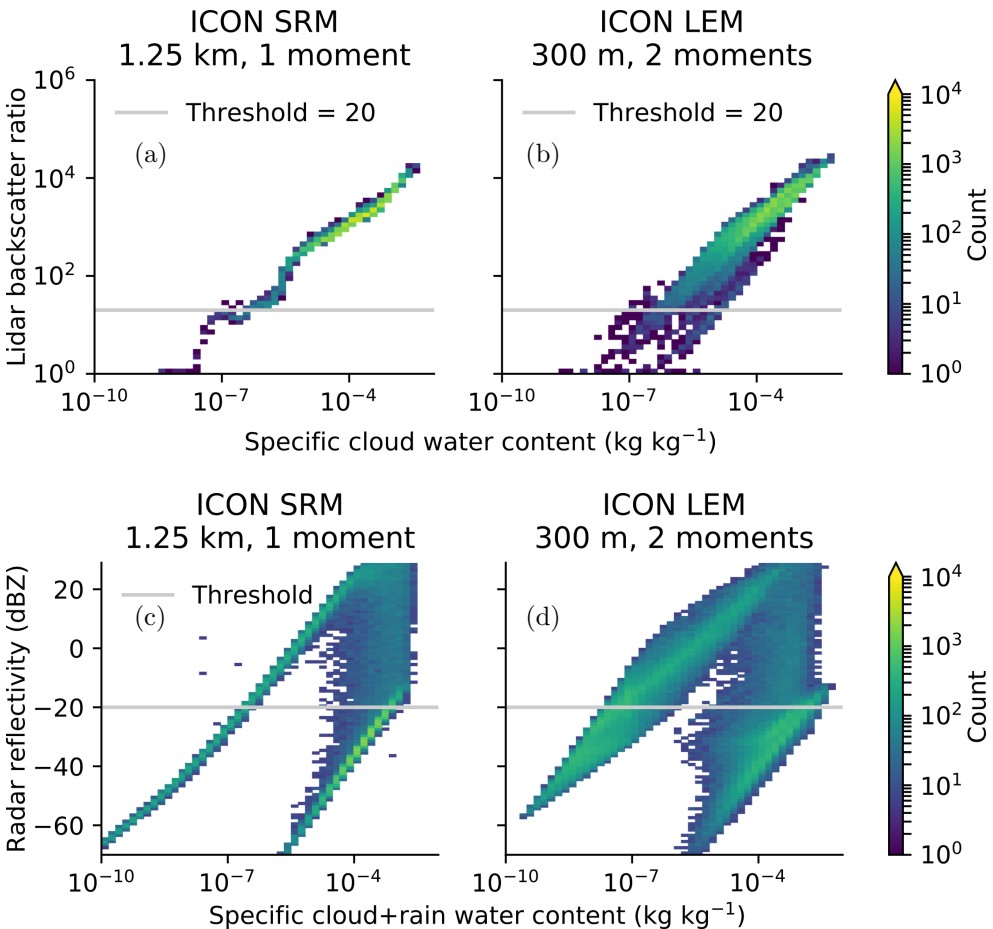

**Figure 2.** Simulated lidar and radar signals as a function of hydrometeor contents. CR-SIM and PAMTRA simulate the observable lidar and radar signals from drop size distributions in the one-moment ICON SRM and two-moment ICON LEM microphysical models. Signals are simulated without attenuation as they would be sensed at cloud top.

## 5.1 Case study

An example scene observed from HALO during research flight 5 is depicted in Fig. 4a. Here, several very shallow clouds close to the LCL were observed first, followed by a precipitating cloud with stratiform shallow anvil outflow. The shallow clouds were only detected by the lidar, whereas the precipitating cloud was detected by both the lidar and the radar. However, the lidar detected cloud top heights about 50 to 100 m, i.e., up to three radar range gates, above the uppermost recorded radar echo. Also, a larger part of the stratiform layer was visible to the lidar. Thus, we conclude that the precipitating shallow cumulus has a thin layer of very small droplets on top which are only seen by the lidar due to its higher sensitivity (compare Fig. 2).

A joint standard grid for the radar and lidar observations and forward simulations is used to facilitate additional analysis. A grid spacing of seven radar range gates is chosen, so that histograms are calculated as counts in 210 m high bins normalized by

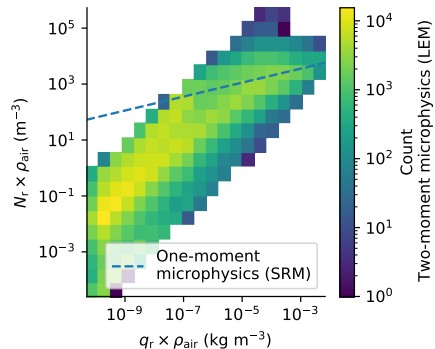

**Figure 3.** Relation of volumetric raindrop number concentration and mass mixing ratio for one- and two-moment microphysics.

the bin width and the total number of cases in the total dataset. The histogram statistics in the right part of Fig. 4a summarize
the detected cloud layers in the scene. The integral over the histogram equals the 2D shallow-cloud cloud coverage detected by
the respective sensor. In the particular scene depicted in Fig. 4, the lidar sees a cloud in about 73 % of the time, while the radar
cloud coverage is about 46 %. Note that the histograms depict the vertical distribution of detected cloud tops or base heights in
a column and are therefore different from profiles of vertical cloud fraction. In the case of multi-layer clouds, individual layers
could be hidden due to attenuation. Therefore only the uppermost cloud top and lowest base are considered. The histogram in
Fig. 4a reveals the separation of the radar echo base into large non-raining droplets in the stratiform layer and precipitation that
falls out of the cloud base at LCL. Note that the lowest usable radar range bin is at about 100 m above the sea surface to avoid
any surface clutter artifacts.

Figure 4b displays an example time series from ICON LEM which also includes precipitating clouds (beginning of the time
series) and a few very shallow thermal-driven clouds (in the end). The cloud tops seen by the lidar and radar are mostly in the
stratiform mode about 2 km above the LCL. The peak of the radar cloud top heights is about 400 m above most of the lidar
cloud tops. This order is contrary to the observed case study and probably caused by evaporation of cloud droplets at cloud top
as the higher reaching radar signal originates from grid cells at cloud top containing only rainwater but no cloud water. This
can be seen by the pixels with a radar reflectivity signal above the lidar cloud top height, e.g., at 20:11. Here, raindrops might
be transported out of the cloud core by wind sheer or turbulence. As only a few thin lidar-only-visible clouds near LCL are
present in this scene, the mode of very shallow clouds is not much pronounced in this example.

## 5.2 Cloud statistics

After introducing and discussing the approach in a case study, all observations and simulations are jointly analyzed in this
section. The histograms of the observed lidar cloud top heights (Fig. 5) reveal two modes of cloud top heights, similar to the
case study. While the lower mode of very shallow clouds is centered at about 300 m above LCL, the stratiform mode is centered
at about 1.3 km above LCL. Frequency wise, stratiform clouds (and active cloud cores included in the stratiform mode) were

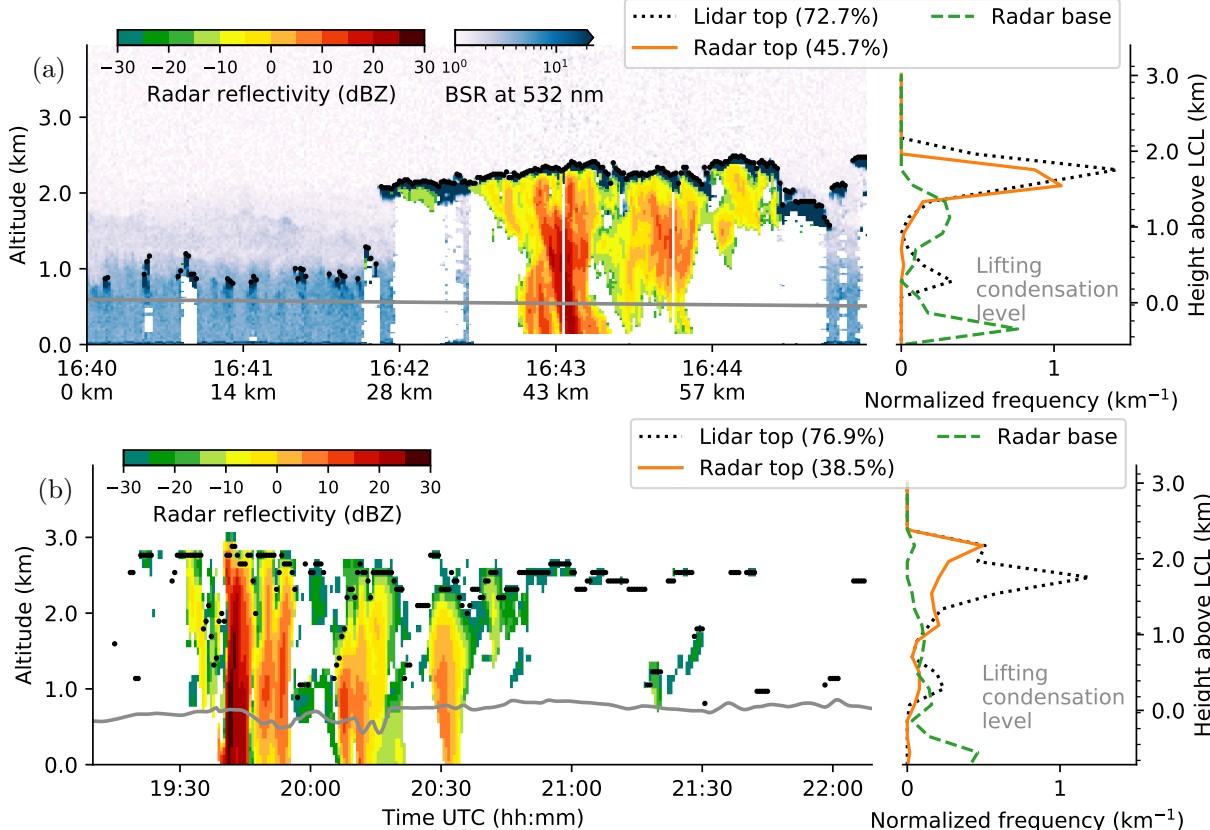

**Figure 4.** Case study time series of observed (a) and modeled (b) radar reflectivity, lidar-detectable cloud top height, lifting condensation level (LCL) and their vertical distribution. Shallow 2D cloud coverages detected by lidar and radar in each dataset are given in each legend. Observations (a) are from research flight 5 on 2013-12-15 and also include lidar backscatter ratio (BSR) plotted below the reflectivity. Model simulation is from an ICON LEM meteogram station on 2013-12-16. The vertical distributions are normalized by the number of time steps in each scene.

observed more than twice as often as very shallow clouds, when they were not hidden beneath such a stratiform cloud. This is contrary to the ground-based impression from the same region but other period when the shallow mode was clearly dominating (Nuijens et al., 2014). The lower mode of very shallow cumulus clouds on top of the well mixed boundary layer (Stevens et al., 2017) is very likely to be thermal driven and hardly produces any precipitation. The radar, however, observes in principle just one mode of top heights with its maximum at about 1.3 km above LCL, consistent with the upper lidar mode. But, similarly to the example in Fig. 4, the distribution is shifted slightly towards lower top heights than the lidar-visible cloud top distribution. Overall, the lidar sees clouds more than twice as often as the radar (43.2 vs. 18.2 %) due to its higher sensitivity that even responds to low cloud water contents of about $10^{-7}\,\mathrm{kg\,kg^{-1}}$ (compare Fig. 2).

We attribute the stratiform mode to shallow convection, precipitating clouds and their shallow anvil outflow. This interpretation is supported by the distribution of radar echo bases. These bases are also bimodal with the upper mode about 400 m below

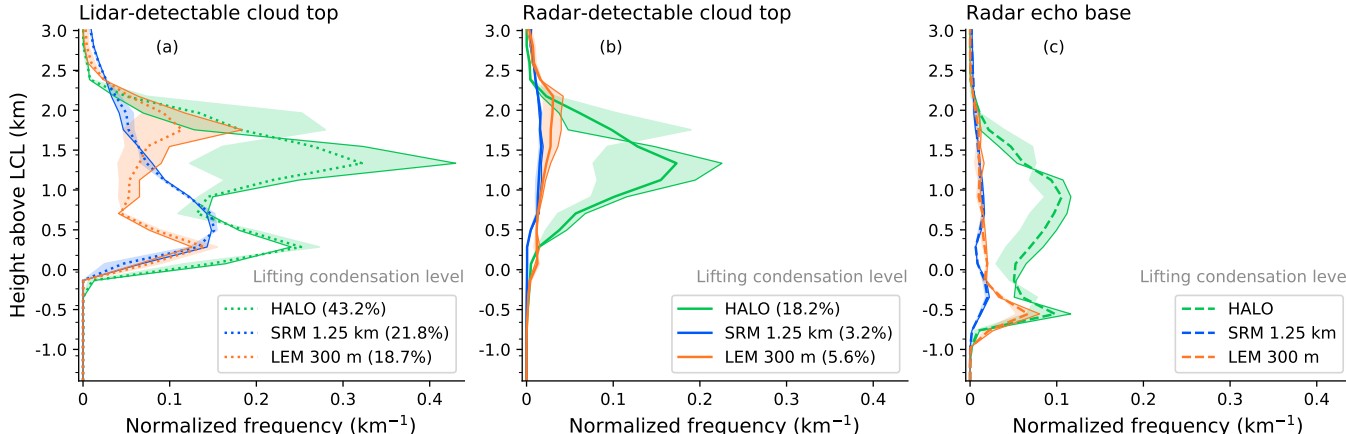

**Figure 5.** Cloud boundary statistics on all observed and forward-simulated lidar and radar signals: (a) lidar cloud top, (b) radar cloud top, and (c) radar echo base. Same thresholds for cloud detection are used for the observed and simulated lidar and radar signals. Height is in relation to the lifting condensation level (LCL). Shadings depict western (bright edge) and eastern (dark edge) half of each dataset. The histogram bin edges are depicted as ticks on y-axis. Total shallow 2D cloud coverage detected by lidar and radar in each dataset are given in the legends. This total cloud coverage can be derived from a cumulative version of this figure presented in the appendix Fig. C1.

the mode of radar top heights. The upper radar base mode spans over the stratiform and very shallow layers. This upper mode is related to the outflow anvils and not-yet precipitating clouds in which the layer of radar-detectable hydrometeors is only a few hundred meters thick. The lower mode of radar base heights is below the LCL, i.e., in the precipitation layer and comprises clearly precipitating cases even if the precipitation occasionally evaporates before reaching the surface. The radar-detectable
cloud depth distribution can also be seen in Fig. 6 (a) as distance to the main diagonal. The joint histogram of radar base and top height confirms that the clouds observed from HALO are often either about 200 to 400 m thin (along the diagonal) and at about 1 km above the LCL or they precipitate (on the left) with similar cloud top heights.

A deepening of the cumulus cloud layer in accordance with a sea surface temperature increase is expected from the stratocumulus decks in the east tropical Atlantic to the cumulus regime in the west (e.g., Wyant et al., 1997). A temperature increase
of about 2 K from east to west in the flight area motives a separation of our data by longitude. The deepening of the cumulus cloud layer can be seen in the HALO observations as the lidar and radar detect the stratiform mode about 400 m higher in the observations west of 51.5° W than east of it. This deepening probably caused the frequency reduction of the stratiform mode in the western half compared to the shallower eastern half. Such relation between deepening of the cloud layer and reduced formation of stratiform clouds was also shown in an LES study by Vogel et al. (2020). However, the relation seems opposite to
the positive correlation between thin stratocumulus cloud fraction and planetary boundary layer depth observed with satellites on monthly timescales in the marine stratocumulus to cumulus transition by O et al. (2018). In contrast to the deeper and stratiform clouds, the frequency and height of very shallow lidar-visible clouds is almost the same in the western and eastern parts, which also agrees with Vogel et al. (2020).

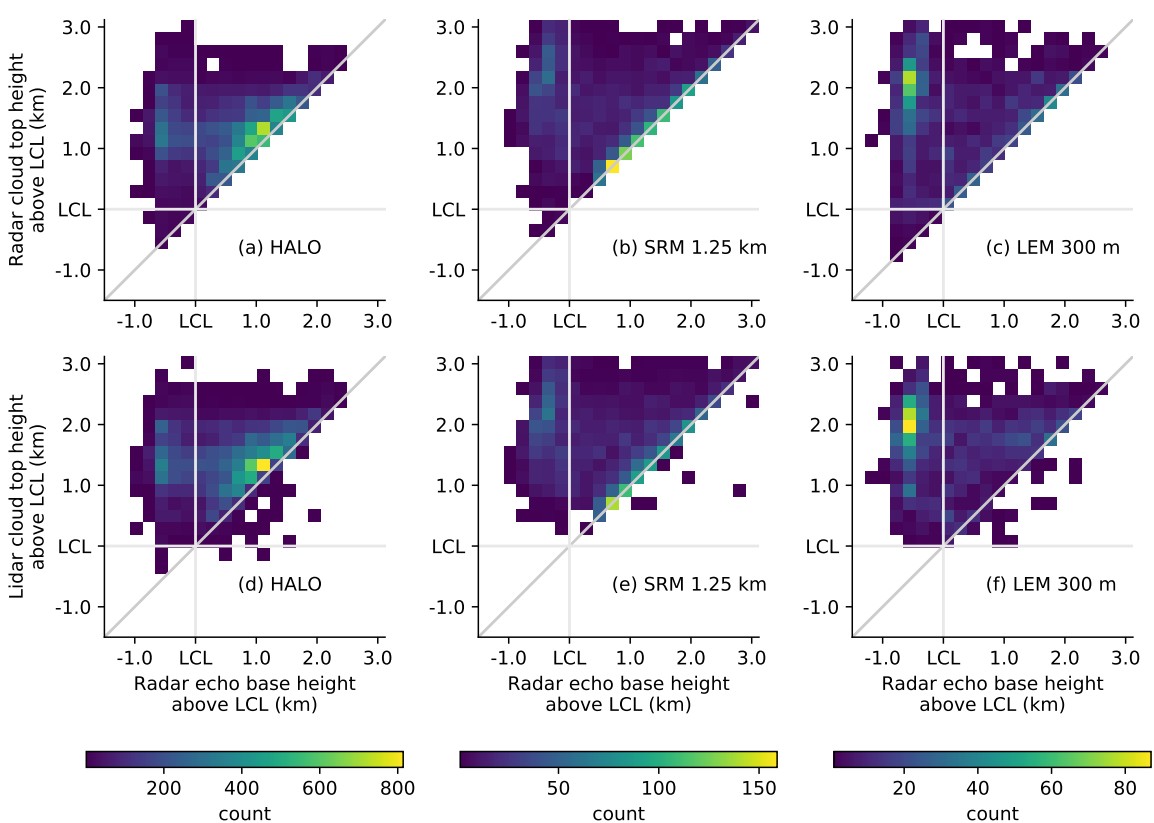

**Figure 6.** Relation of radar-detectable cloud top height (a-c) and lidar-detectable cloud top height (d-f) above LCL versus radar echo base height above LCL. Based on HALO observations (a, d), 1.25 km SRM dataset (b, e), and 300 m LEM dataset (c, f).

More pronounced than in the case study, a bimodal distribution of cloud top heights is also present in the dataset of all available ICON LEM output (Fig. 5). The mode of very shallow clouds behaves very similar to the observations. It has its maximum frequency at the same height and is also only detectable with the sensitivity of the lidar. However, the frequency of this mode and the overall total cumulus cloud coverage is only half the observed cloud coverage (18.7 vs. 43.2 %). The height of stratiform mode is about 400 m above the observed stratiform mode of the whole HALO dataset, but matches the stratiform mode of the western part of observations very well. This is in line with the fact that the LEM is only represented by meteograms in the western flight area. Clouds are detected by the forward-simulated radar in only 5.6 % of the LEM scenes compared to 18.2 % in the observations. In agreement with the observations, the radar cloud tops are mostly modeled in the stratiform layer of the LEM, but with the maximum higher than the lidar cloud tops – similar to the example discussed before. The distribution of the modeled radar signal base heights indicates that most clouds in the LEM are precipitating if they are visible to the radar. In contrast to the observations, non-precipitating radar-detectable clouds in the LEM develop with a wider range of depths as shown in Fig. 6 (c).

The ICON SRM represents the clouds rather differently than the LEM. The clouds visible to the lidar generally form one broad mode with the most frequent lidar cloud top heights around 500 to 700 m above LCL. The frequency of cloud tops decreases with altitude until they disappear at 2.6 km above LCL, which is similar to the other two datasets. The clear separation of cloud tops into two layers, however, is not evident in contrast to the observations and the LEM. While a double layer structure

could be seen on individual days in the SRM data (not shown), it does not show up on average due to the strong variation in altitude between different days. Radar-detectable clouds and precipitation are also modeled but only in about 3 % of the SRM scenes, which is much less than observed (18 %) and in the LEM (6 %). The radar top height distribution, however, has a similar shape compared to the observed radar clouds. Even if less frequent, the relative distribution of radar signal base heights in the SRM is similar to the observations with one peak between LCL and LCL + 1 km (spanning over the stratiform and

very shallow layers) and the second peak few hundred meters below LCL in the precipitation layer. Like in the observations, there is a similar mode of thin radar-detectable clouds (Fig. 6b, along the main diagonal), however, at generally lower heights, and precipitating clouds with deeper cloud tops compared to the observations. The clear difference of observed stratiform and precipitating cloud layer heights between the eastern and western part of the data is not pronounced in the SRM data, even though the coverage of the SRM matches the observations better than the LEM. This indicates that the shallow convection and

outflow process as well as the formation of a stratiform layer are modeled differently than observed.

There are a couple of observed cases (right of the diagonal in Figures 6 d-f) with the lidar-detected cloud top heights below the radar base height. These signals relate to lateral raindrop transport out of the precipitating core with a patch of cloud beneath. Such cases also occur in the LEM dataset and less frequently in the SRM dataset. The smaller grid spacing in the LEM could be favorable for a more likely lateral transport of raindrops into a neighbor grid cell in the LEM compared to the

350 SRM.

We can compare the observed and modeled bimodal cloud top distribution to the those found by Genkova et al. (2007) and Leahy et al. (2012). The modes identified by the former at 650 and 1500 m above sea level seem to be lower than the ones observed in the present study with the LCL height at about $750 \pm 150$ m, even though they studied a similar domain in a similar season. However, Genkova et al. (2007) denote a vertical uncertainty of 250 to 500 m. The modes observed in similar

cloud regimes by Leahy et al. (2012) at around 800 and 2000 m are closer to the values in the present study, even though the observations took place in a different domain over the tropical South Pacific.

To conclude: Bimodal lidar cloud top height distributions were observed and their clear separation is well reproduced by the LEM but not by the SRM. The lower mode of very shallow thermal-driven clouds is closely above the LCL, while the upper mode is closely below the trade inversion (Stevens et al., 2017), i.e., about 1.3 km higher up. The SRM, however, shows

one prominent mode of cloud top heights with its maximum at rather lower heights. However, the SRM also produces deeper clouds with their frequency decreasing with height. Neither model reproduces the often observed radar echoes embedded in the non-precipitating upper stratiform mode. To shed light on the conditions under which these clouds are underrepresented compared to observations, comprehensive LWP observations refine the statistics in the next section.

## 5.3 LWP classes

The stratification of the observations and model output into different LWP classes can give more detailed insight into the regimes under which the models perform better or worse. LWP classes are chosen to represent barely detectable clouds ($< 10\,\mathrm{g\,m^{-2}}$), clouds which are not completely optical thick ($< 50\,\mathrm{g\,m^{-2}}$), classical cumulus clouds ($50\,\mathrm{g\,m^{-2}} < \mathrm{LWP} < 100\,\mathrm{g\,m^{-2}}$), thicker clouds which are still considered in satellite retrievals ($100\,\mathrm{g\,m^{-2}} < \mathrm{LWP} < 300\,\mathrm{g\,m^{-2}}$) (Wentz and Meissner, 2000), and even more water bearing clouds ($300\,\mathrm{g\,m^{-2}} < \mathrm{LWP} < 1000\,\mathrm{g\,m^{-2}}$). An overview of cloud top heights and radar base in the

different datasets and LWP ranges is presented in Fig. 7 and discussed in the following.

It is remarkable that high cloud tops in the stratiform layer were often observed by the lidar under low LWP conditions (below $10\,\mathrm{g\,m^{-2}}$). Such clouds likely correspond to thin "veil" clouds frequently observed near the upper boundary layer, i.e., below the trade inversion, in the stratocumulus to cumulus transition by Wood et al. (2018) and O et al. (2018). They report on geometrically and optically thin clouds with low droplet number concentration (about $5\,\mathrm{cm^{-3}}$) but relatively large droplets

with radii ranging from 15 to $30\,\mu\mathrm{m}$. Droplets of such sizes are large enough to provide a radar reflectivity above the detection threshold.

Extending the LWP class from $10\,\mathrm{g\,m^{-2}}$ to $50\,\mathrm{g\,m^{-2}}$ includes more additional lidar-detectable stratiform cloud coverage to the statistics than very shallow cloud coverage. This means even more veil clouds are included, which were estimated to have a typical LWP of about $25\,\mathrm{g\,m^{-2}}$ (Wood et al., 2018). In all cases with LWP $< 50\,\mathrm{g\,m^{-2}}$, the stratiform layer was observed about

380 1.5 times more often by the lidar than the layer of very shallow clouds, which is a bit more often than in the LEM and SRM (see also Fig. 8a).

In general, it is no surprise that the distributions of lidar cloud tops in low LWP conditions ($< 50\,\mathrm{g\,m^{-2}}$, Fig. 7a and d) are similar to those of the whole dataset (Fig. 5a), as most of the scenes have a low LWP (compare Fig. 11b in Jacob et al. (2019) for the HALO dataset). The statistics of radar-detectable cloud top and base heights in scenes with LWP $< 10$ and $50\,\mathrm{g\,m^{-2}}$ in

Figs. 7b, c, e, and f are different from the overall statistics (Fig. 5b), as the radar is often not sensitive enough to detect clouds with such little LWP. The lidar-detected clouds are about seven (three) times more frequent than those detected by the radar on HALO in scenes with LWP $< 10\,\mathrm{g\,m^{-2}}$ ($< 50\,\mathrm{g\,m^{-2}}$). In the LEM simulations, this ratio is about five for both LWP limits. The relative smaller increase in radar-detectable clouds means that clouds in the LEM with $10 < \mathrm{LWP} < 50\,\mathrm{g\,m^{-2}}$ probably have only small droplets and thus miss a radar-detectable drizzle component. About a twelfth of the observed radar clouds with

LWP $< 50\,\mathrm{g\,m^{-2}}$ are categorized as precipitating, while the LEM depicts half of them as precipitating. No statement on the SRM precipitation fraction can be made as only $0.2\,\%$ (i.e., less than 200 profiles) of the SRM scenes with LWP $< 50\,\mathrm{g\,m^{-2}}$ show radar-visible cloud tops below 4 km at all.

The lidar detected a cloud in $96\,\%$ of the observed scenes with LWP $> 50\,\mathrm{g\,m^{-2}}$. In the remaining cases, the lidar either missed clouds with only partial coverage in the microwave radiometer footprint ($\approx 1\,\mathrm{km}$) or few but large raindrops were horizontally

transported out of the cloud core, such that they are only visible to the microwave radiometer. Likewise, not all clouds in scenes with LWP $> 50\,\mathrm{g\,m^{-2}}$ contained radar-detectable hydrometeors. This difference between lidar- and radar-detectable clouds with LWP $> 50\,\mathrm{g\,m^{-2}}$ is in principle also reproduced by both models. In the observations, about four of five clouds detected by the

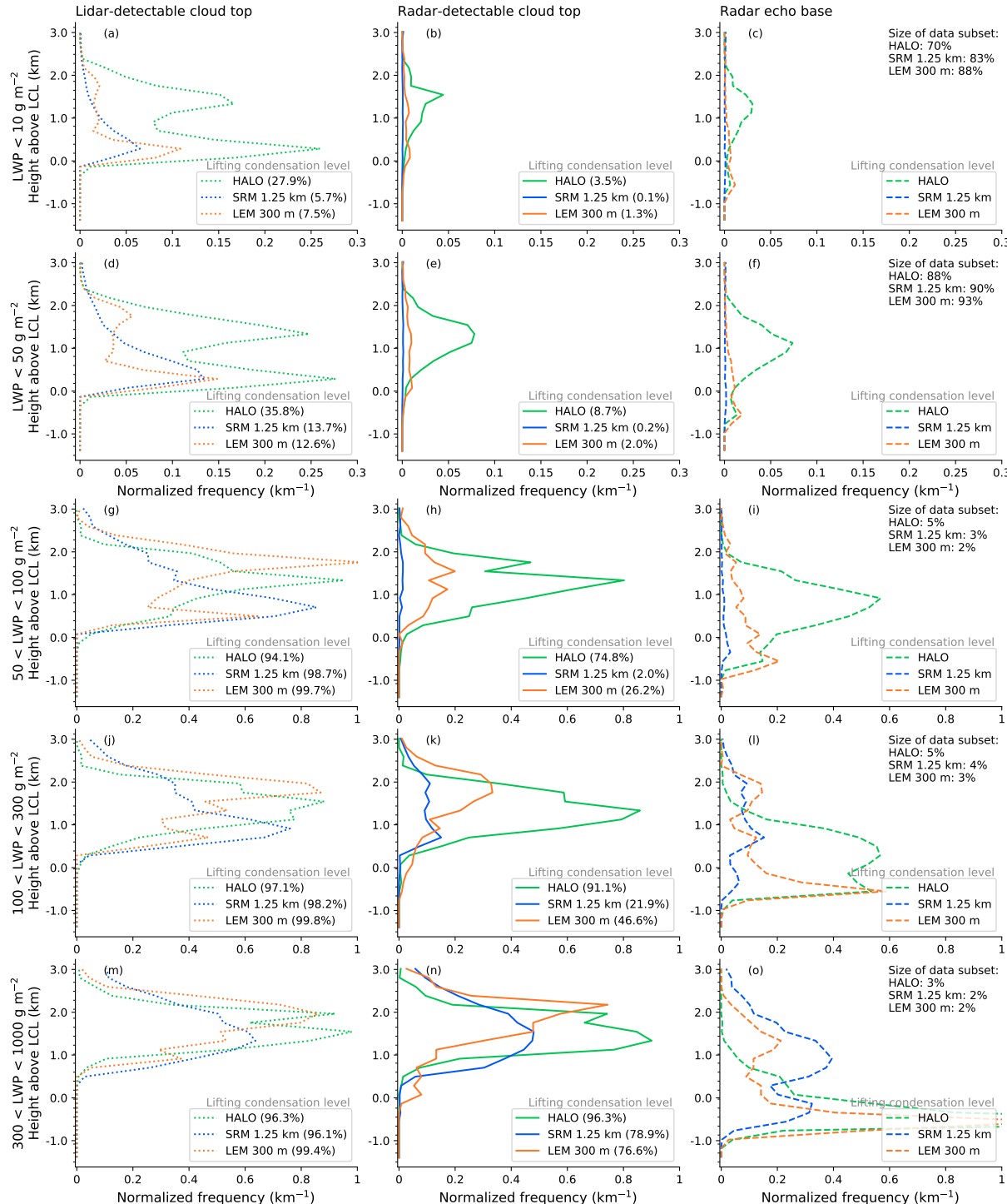

**Figure 7.** Similar to Fig. 5 but classified by liquid water path (LWP). Columns represent lidar cloud top, radar cloud top, and radar base of observed and forward-simulated lidar and radar signals. Rows represent different LWP ranges. Note the different x-scale used in the upper two rows.

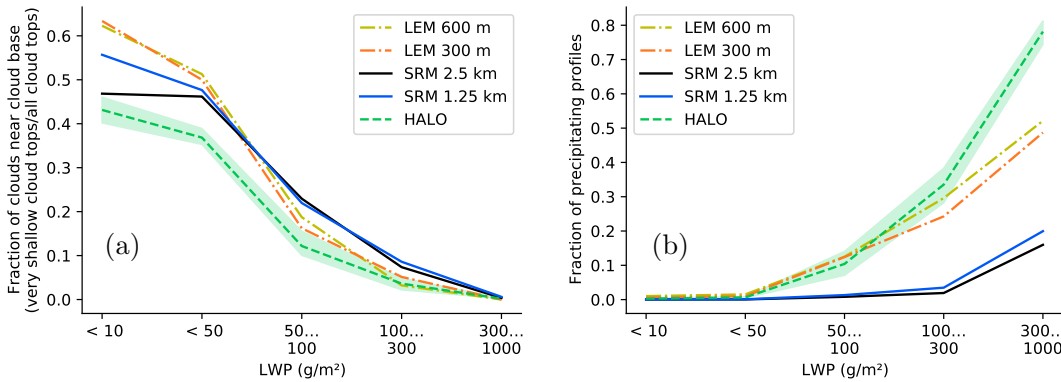

**Figure 8.** Fraction of very shallow clouds and precipitating clouds as a function of LWP. (a) Fraction of clouds near cloud base defined as ratio of very shallow clouds (lidar-detectable cloud top within LCL and LCL + 600 m) to all shallow clouds (lidar-detectable cloud top height < 4000 m). (b) Fraction of precipitating profiles defined as ration of radar echo below LCL to all profiles. Shading of the HALO dataset represents the uncertainty of the LWP retrieval.

lidar were also seen by the radar in the 50 to 100 g m$^{-2}$ LWP class. However, only a quarter of the lidar-detectable LEM clouds are also detectable by the simulated radar. The ratio in the SRM simulations is even smaller. The radar base on the other hand

shows that the LEM models about half of the radar-detectable clouds as precipitating, while precipitation was only observed for a quarter of the observed radar clouds with $50 < LWP < 100$ g m$^{-2}$ (compare Fig. 6).

In scenes with LWP between 100 and 1000 g m$^{-2}$, the radar-detectable clouds in both models form two groups. They either precipitate or belong to the stratiform layer with a base clearly above the LCL. Such a separation was not observed from HALO. The mode of non-precipitating radar-visible clouds under high LWP ($> 300$ g m$^{-2}$) conditions in both models, however, can be

explained by heavy clouds in the model consisting purely of cloud droplets. A model cloud with LWP $> 300$ g m$^{-2}$, for example, which is 300 m deep must on average contain at least $10^{-4}$ kg kg$^{-1}$ liquid. This means that such a cloud is radar-visible without containing any raindrops (compare Fig. 2).

Figure 8 summarizes the dependence of the very shallow cloud mode and precipitation on the LWP classes. It includes the output from a coarser nest of both ICON model versions to shed light on the influence of grid-resolution versus model

configuration. In general, all models and resolutions simulate the observed and expected reduction of very shallow clouds with increasing LWP (Fig. 8a). However, relatively more very shallow clouds were modeled than observed. The likelihood of precipitation with higher LWP (Fig. 8b) and hence also in deeper clouds, separates the datasets. The increase in precipitation up to LWP $\leq 300$ g m$^{-2}$ in both LEM resolutions corresponds better to the observations than the SRM outputs do. For very high LWP, too little high-LWP clouds precipitate, which could be caused by a too weak autoconversion process in the LEM. The

additional outputs from coarser resolved LEM and SRM nests suggest the different resolutions being less important than the choice of parametrizations, vertical levels and microphysical scheme. Differences in sampling of the LEM and SRM outputs, however, also influence the results, but sensitivity testing the sampling area and considered dates showed no major influence as analyzed in appendix C.

The stratification of the data by LWP shows that both models cannot represent non-precipitating but radar-visible drops that were observed under all and especially low LWP conditions. These drops are probably larger than those represented by the Gamma distributions of the cloud hydrometeor class in both models. Radar-visible model clouds precipitate more often than observed, which means they contain already very large droplets, but the fraction of radar-visible clouds is in general too small. Non-precipitating clouds, consisting of cloud-type hydrometeors only were produced by both models under high LWP conditions ($> 300 \, \mathrm{g \, m^{-2}}$), but such cases were not observed.

## 6   Summary and conclusions

Observed statistics of hydrometeor profiles and liquid water path (LWP) of oceanic shallow cumulus clouds are compared against those produced by two high resolution models of the ICON family. The observations and model runs were part of the NARVAL experiment over the tropical Atlantic east of Barbados in the dry winter season 2013. The instruments were operated from the research aircraft HALO at an altitude between 13 and 14.4 km in a nadir-pointing orientation. The two models are the so-called storm resolving model (SRM) and the large-eddy model (LEM). Primarily, outputs with grid-spacings of 1.25 km (SRM) and 300 m (LEM) are analyzed, but additional outputs are coarser grid-spacings (2.5 km and 600 m) are considered as well. The SRM resolves the shallow cumulus layer (700 – 3000 m above sea level, (asl.)) with 14 levels, while the LEM has 24 levels in that layer.

The upper part of the hydrometeor profile is characterized by radar and lidar observations, while the lower part of the hydrometeor profile is characterized by the radar only. The LWP is retrieved from microwave radiometer measurements. When looking at the high occurrence of low-LWP scenes in the models (83 and 88 % below $10 \, \mathrm{g \, m^{-2}}$ in SRM and LEM, respectively, Fig. 7), it becomes evident that common sensitivity thresholds for the instruments and models are urgently needed to assess clouds in this regime. Thus, forward simulations of the radar and lidar observations using instrument specific sensitivity thresholds and relationships between the observables and the model output are used to allow an apples-to-apples comparison between the HALO observations and the ICON model output (Lamer et al., 2018). A lidar backscatter ratio threshold of 20 suggested by Gutleben et al. (2019) is applied to clearly distinguish between backscatter from dust aerosols and cloud droplets. A comparison of the airborne measurements to ground-based radar records reveals a reliable radar reflectivity detection threshold of $-20 \, \mathrm{dBZ}$ for the airborne radar over the full column. The forward simulations show that most clouds with $q_\mathrm{c} > 10^{-7} \, \mathrm{kg \, kg^{-1}}$ in the model are detectable with the respective backscatter lidar threshold. The radar, in contrast, is primarily sensitive to the "rain"-category hydrometeors in ICON. Only the highest amounts of liquid $q_\mathrm{c}$ in a cloud-water-only cloud in the model are detectable by the radar.

The observations reveal a bimodal distribution of cumulus cloud top heights separating the clouds into a mode of very shallow cumulus, defined from the lifting condensation level (LCL) to LCL + 600 m, and a stratiform layer, defined from LCL + 600 m to 4 km asl.. The very shallow mode of cloud tops relates to shallow, non-precipitating boundary layer clouds reaching up to a few hundred meters above the LCL. The stratiform mode is mostly driven by shallow moist convection and also contains detrainment clouds, which are often formed from the outflow of active shallow cloud anvils at around 1.3 km above LCL. The

very shallow mode consists of mostly thin water clouds that are best seen by the backscatter lidar and are frequently missed by the radar. In contrast, clouds in the stratiform mode contain more and larger droplets that scatter sufficient microwave radiation to be detected by the radar in addition to the lidar. Overall, the stratiform mode was observed 2.5 times as often as the very shallow mode and was in general more frequent in scenes with higher LWP ($> 50\,\mathrm{g\,m^{-2}}$) than in those with lower LWP. In the stratiform layer, the lidar detected the cloud tops slightly higher than the radar. This indicates that small particles with low radar reflectivity are present at the upper part of the stratiform layer. Higher LWP values are associated with more precipitation echoes below the LCL and with deeper clouds, even though already 10 % of the scenes with medium LWP ($50 - 100\,\mathrm{g\,m^{-2}}$) showed precipitation. Also, a clear trend with higher cloud tops in the stratiform mode in the western part of flight tracks is observed that is probably related to higher sea surface temperatures in that area enforcing convection.

The bimodal cloud top height distribution is reproduced by the LEM, although the total cloud fraction is lower than observed. The radar forward simulations show that too few large particles or in general too small cloud droplets in the stratiform regime are produced in LEM. The observed increase of radar-detectable clouds between LWP of 10 and $50\,\mathrm{g\,m^{-2}}$ is not reproduced by the LEM. This is consistent with the overall trend of the models that produce to few large particles. However, the LEM describes more of the radar-detectable clouds as precipitating than observed. This indicates that large radar-visible drops probably cannot be kept long enough in the model cloud layer before falling out. An observed cloud layer deepening with LWP can be also found in the LEM.

Different than the LEM, the SRM produces no clear separation between the two cloud layers. Cloud tops are typically at 500 to 700 m above LCL. Small differences in the warm autoconversion (AU) parametrizations might be a reason for the reduced frequency of deeper shallow clouds. The AU formulation is similar in the LEM and the SRM, but as the SRM cloud droplet number concentration $N_c$ is constant (Table 1) but smaller than the average in-cloud $N_c$ in the LEM, and as the AU rate increases with decreasing $N_c$ (Seifert and Beheng, 2001, eq. 16), the AU in the SRM is expected to be stronger on average. Therefore rain could form quicker in the SRM and thereby reduce the average cloud life time, cloudiness, and also cloud top height. Indeed, especially the radar-visible cloud top heights of the LWP heavy clouds in the SRM are in general lower than in the LEM (Figs. 7k and 7n). One could hypothesize further that a faster warm-precipitation-cycle reduces the strength of the shallow convection, such that fewer clouds reach the trade inversion, which would force the cloud to create the shallow outflow in the stratiform layer, that is produced too rarely by the SRM. However, there are other differences between the LEM and SRM that could contribute to differences in cloudiness and rain production. For example, the lack of a clear gap in the cloud top frequency distribution might be also due to the lower vertical resolution of the SRM which would require always the same few model layers to be cloud free. Furthermore, higher horizontal resolution influences the cloud formation and resolves buoyancy production better. However, the models and model outputs used here indicate a stronger dependence of cloud production on the model setup than on the resolution as shown in appendix C. The clearly observed east-west difference in the height of the stratiform cloud layer is only weak in the SRM. This indicates that processes of the precipitating shallow-convection cumulus clouds are not fully represented in the SRM. The SRM cloud distribution is rather insensitive for different LWP classes except for a cloud deepening and precipitation increase with increasing LWP. This study primarily considers the grid-resolved clouds in the SRM. This might be an unfair comparison as the SRM also contains a diagnostic scheme for sub-grid-scale cloudiness

used in the radiation calculations. Thus, the additional sub-grid-scale cloudiness is briefly assessed in the appendix B. In summary, clouds modeled from diagnostic equations would moderately increase the SRM cloudiness, but would not alter the vertical structure significantly, i.e., the diagnosis does not solve the missing cloudiness in the stratiform layer.

Both models show clearly non-precipitating radar-visible clouds with LWP > 300 g m$^{-2}$ which were not observed in that way and probably come from very high amounts of pure cloud water. In other cases, both models tend to produce precipitation that is also detectable below LCL once the cloud is visible to the radar and it seems that large radar-visible but only slowly-sedimenting non-precipitating drops like in drizzle are missing. This is probably due to the size constraint in the ICON microphysics (Seifert and Beheng, 2001), that implies a threshold between cloud PSD and rain DSP at 40 μm, i.e., cloud PSD is assumed to not

contain a significant number of droplets with diameter larger than this threshold. Our observation of larger but non-precipitating particles is in line with findings by Siebert et al. (2013) and Wolf et al. (2019) who observed cloud droplet effective radii on the order of this threshold in the same region but in generally moister months, i.e., they also note the principle presence of large cloud droplets.

Finally, it has to be noted that the available datasets have a great spatiotemporal overlap but do not match perfectly. The

consequences of this are probably less severe than they would be for example in the mid-latitudes, a region that is heavily influenced by synoptic systems, because the study area and period is characterized as mostly undisturbed (Vial et al., 2019) and the variation from flight to flight in the winter season is limited (Jacob et al., 2019). The exact choice of domain and dates used in this study are analyzed in the appendix C by taking subsamples from the 1.25 km SRM dataset, but no significant impact on the cloud statistics can be found. The methods presented in this study show high potential to benchmark realistically driven

large-eddy simulations. Even if the matching between model and observations could be improved in future studies the analysis provides insight into processes that are well represented by the models and which phenomena are difficult to model with the respective setup.

Enhanced observations with several research aircraft, vessels, and autonomous platforms and coordinated model applications during the recent EUREC[4]A field study in early 2020 (ElUcidating the Role of Cloud-Circulation Coupling in Climate Bony

et al., 2017) will provide an even more comprehensive view on the trade wind shallow cumulus clouds. For that, the methods presented here are ready to by applied to future EUREC[4]A studies. Also, cloud-chasing ship-based observations can observe individual cloud cycles including the transition from pure cloud to drizzle onset and probably rain production, while airborne observations survey the cloud field to report on the representativeness of the in-detail studied cloud. As shallow cumulus clouds also were be probed in-situ in addition to the remote sensing setup used in this study, a closer look into the drop size

distributions in the stratiform layers will be enabled in upcoming studies.

*Code and data availability.* The source code of CR-SIM was made available by Oue et al. (2020) at https://www.bnl.gov/CMAS/cr-sim.php (last accessed online: Nov. 6, 2019). The PAMTRA source code was made available by Mech et al. (2020) at https://github.com/igmk/pamtra/ (last accessed online: Nov. 6, 2019). The airborne radar and dropsonde can be found under https://doi.org/10.1594/WDCC/HALO_measurements_2 (Konow et al., 2019). The LWP retrieval data from the HAMP microwave radiometer can be found under https://doi.org/10.26050/WDCC/HALO_measurer

(Jacob et al., 2019). The BCO data are accessible to the broader community through Stevens et al. (2015). The ICON SRM and LEM outputs were produced by Klocke et al. (2017) and made further public by Vial et al. (2019) and Stevens et al. (2019). The set of ICON model output used in this study is available at the long-term archive of the German Climate Computing Center (DKRZ; dataset DKRZ_LTA_834_ds00052 at https://cera-www.dkrz.de/WDCC/ui/cerasearch/entry?acronym=DKRZ_LTA_834_ds00052).

## Appendix A: Radar sensitivity

To estimate the practical sensitivity limit of the HALO radar observations, HALO radar statistics are compared to ground-based measurements taken at the BCO. The BCO radar operates at the same Ka-band frequency as the airborne radar, but has a better sensitivity due to a larger antenna and longer integration time (Lamer et al., 2015). Therefore, the lower MDS of the BCO radar offers the opportunity to assess the practical sensitivity limit of the HALO radar.

A comparison can only be made on a statistical basis as the BCO and HALO radars do not sample the same volume. To
530 avoid statistical effects of the diurnal cycle identified by Vial et al. (2019), BCO data are only considered roughly during the time when HALO was flying, i.e., between 12:00 and 21:00 UTC (8:00 and 17:00 local time) on the 8 flight days.

The higher BCO radar sensitivity compared to the HALO radar is notable in the height-resolved reflectivity histograms in Fig. A1. The BCO radar frequently measures reflectivity signals down to $-70\,\mathrm{dBZ}$ at around $500\,\mathrm{m}$ with a clear frequency maximum below $1\,\mathrm{km}$ for $Z$ up to $-20\,\mathrm{dBZ}$. Klingebiel et al. (2019) identify such weak signals at BCO below $-50\,\mathrm{dBZ}$ as
originating from sea salt aerosols and only signals above $-50\,\mathrm{dBZ}$ are attributed to clouds. Clouds with reflectivity between the HALO radar MDS ($-32\,\mathrm{dBZ}$) and $-20\,\mathrm{dBZ}$ and within $4\,\mathrm{km}$ above sea level are observed in $8.5\,\%$ of the time at BCO but only rarely ($< 1.2\,\%$) by HALO. Only clouds with a reflectivity higher than about $-20\,\mathrm{dBZ}$ are similarly or more often observed by HALO than at BCO. Thus, we use $-20\,\mathrm{dBZ}$ as the practical cloud detection threshold of HALO and use this value to define "radar-detectable clouds" in the observations and forward simulations.

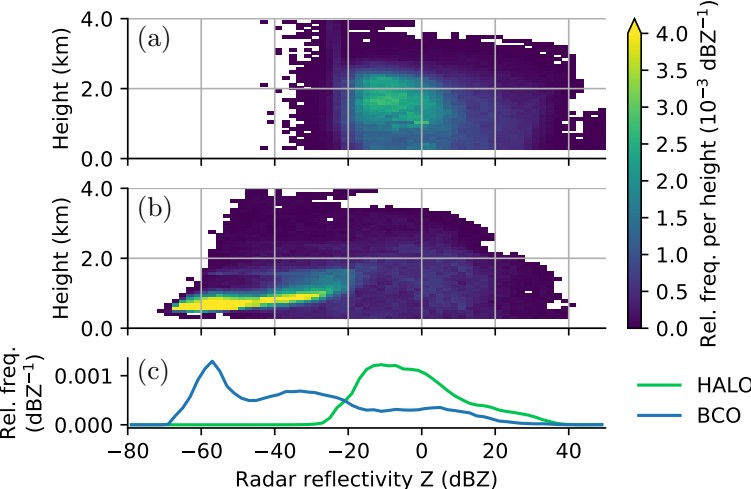

**Figure A1.** Height-resolved radar reflectivity distribution of shallow cumulus from (a) HALO radar and (b) BCO radar during flight days of NARVAL-South. Marginal distributions (c) show the probability density of reflectivity from HALO and BCO below 4 km. BCO data are limited to hours between 12:00 and 21:00 UTC (8:00 and 17:00 local time) on every flight day to match aircraft operation time. The probability density function of each height is normalized to the maximal possible number of data points.

## Appendix B: Sub-grid clouds in the SRM

The vertical cloud structure of the SRM deviates stronger from the observations than that of the LEM, as discussed in Sect. 5. This might be because the forward simulations of the SRM clouds and precipitation are analyzed based on the prognostic model outputs under the assumption that these clouds are resolved by the model grid. However, in addition to the prognostic cloud scheme, the SRM uses a diagnostic cloud scheme to model the sub-grid-scale cloud distribution used in the SRM radiation scheme. This appendix presents a rough estimation, whether the diagnostic cloud scheme could provide the missing clouds in the stratiform mode.

The diagnostic cloud scheme uses a simple box probability density function of total water content and provides the diagnostic cloud fraction (CF) and liquid cloud water content ($q_{c,dia}$) (Martin Köhler, personal communication). In that scheme, the total amount of water is conserved but redistributed between the vapor, and liquid and solid cloud phases. It is assumed, that turbulent perturbations distribute the total water content in a probability density function (PDF) of rectangular shape centered around its prognostic grid-box mean. The supersaturated part of this PDF is then interpreted as diagnostic cloud cover and $q_{c,dia}$. In principle, the diagnostic clouds should be analyzed as filling only their specific CF of each grid box. This means, that the diagnostic in-cloud cloud water $\frac{q_{c,dia}}{CF}$ covers the CF fraction of a grid box. An appropriate analysis of the cloud top height distribution of the sub-grid cloudiness in the SRM would require an assumption on the horizontal overlap of sub-grid clouds within a model column. To circumvent an assumption on this, the influence of the sub-grid clouds is analyzed in terms of the

CF profile. The lidar-detectable $CF_{lidar}$ in each height can then be calculated as

$$CF_{lidar} = \frac{1}{N} \sum_{i=1}^{N} c_i \qquad \text{(B1)}$$

$$c_i = \begin{cases} CF, & \text{if } \frac{q_{c,dia}}{CF} > t \\ 0, & \text{else,} \end{cases} \qquad \text{(B2)}$$

with $N$ being the number of model columns, $i$ the column index, and $t$ the detection threshold. $CF_{lidar}$ describes the spatial
cover in each height that contains enough cloud water to be detectable by the lidar. Analogous to the analysis in the previous section, the prognostic CF is calculated as fraction of cells in one height level, where $q_c > t$. This is a binary assumption that implies full cloud cover, if the cloud simulated from the prognostic equations is lidar detectable.

    The additional CF due to the diagnostic scheme is largest (about 3.5 %) near the LCL (Fig. B1) using the sensitivity threshold $t = 10^{-7}\,\mathrm{kg\,kg^{-1}}$ estimated from Fig. 2a. However, sensitivity tests (not shown) indicated, that the diagnostic and prognostic CF
profiles derived from sensitivity thresholds between $10^{-5}$ and $10^{-8}\,\mathrm{kg\,kg^{-1}}$ are not significantly different. The highest diagnostic CF is at the same height as the prognostic CF at about $500\,\mathrm{m}$ above LCL but about a third higher. Above its maximum, the additional CF decreases until it approaches the prognostic CF. The diagnostic lidar-detectable $CF_{lidar}$ profile follows the profile of diagnostic CF from the model very closely. This means, the lidar is so sensitive, that it detects all (diagnostic) model clouds with meaningful spatial extent.

As the profile shape of diagnostic clouds is very similar to the profile of prognostic clouds, we do not expect the statistics of forward-simulated diagnostic clouds to differ much from what is discussed in Sections 5.2 and 5.3 except for a somewhat higher frequency of lidar-detectable cloud tops. However, a proper forward simulation would have to take the sub-grid cloud overlap problem into account. The radar cloud top and base statistics are almost unaffected by the diagnostic cloud water content, as the maximum additionally diagnosed cloud water content in the SRM is only $2.2 \times 10^{-4}\,\mathrm{kg\,kg^{-1}}$. Such contribution
is insignificant for the radar-detectable cloudiness in relation to the radar detection threshold (compare Fig. 2c).

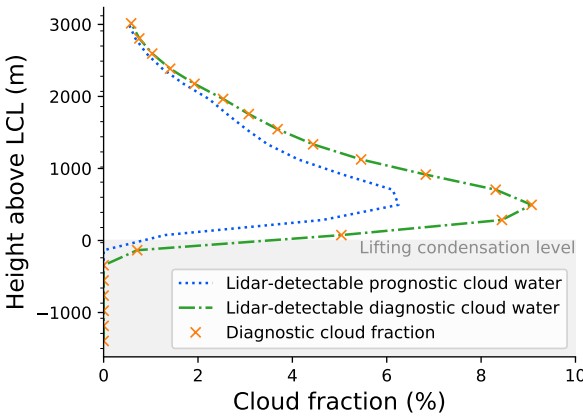

**Figure B1.** Mean cloud fraction profile for resolved and diagnostic lidar-detectable clouds in the full SRM output. Additionally the cloud fraction profile given by the diagnostic equations is shown.

## Appendix C:  Cumulative cloud cover and influence of model resolution and domain

The distribution of cloud top height detected by lidar and radar as well as the radar echo base in the observations and simulated from the model output is shown in Fig. 5. Figure C1 shows a cumulative version of that Fig. 5 to enable the comparison to analogous presentations in the literature (e.g., Medeiros et al., 2010; van Zanten et al., 2011).

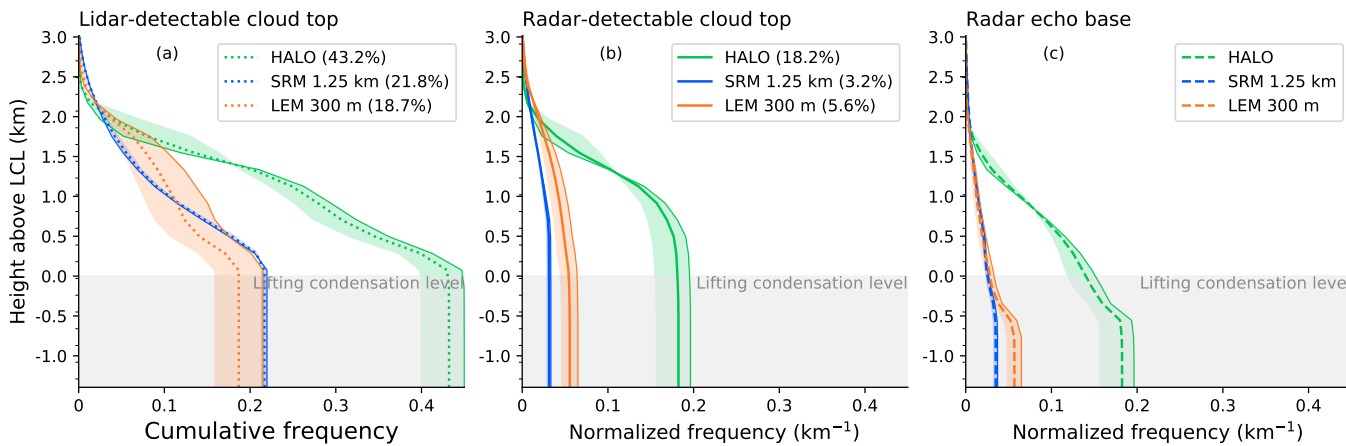

**Figure C1.** Cumulative cloud cover statistics on all observed and forward-simulated lidar and radar signals: (a) lidar cloud top, (b) radar cloud top, and (c) radar echo base. Same thresholds for cloud detection are used for the observed and simulated lidar and radar signals. Height is in relation to the lifting condensation level (LCL). Shadings depict western (bright edge) and eastern (dark edge) half of each dataset. The histogram bin edges are depicted as ticks on y-axis. Total shallow 2D cloud coverage (i.e., the x-axis intercept) detected by lidar and radar in each dataset are given in the legends. This figure is the cumulative version of Fig. 5.

Figure C2 investigates the influence of model resolution by including outputs from two further ICON domains with coarser grid spacings. The figure shows higher similarity among the outputs when refining the horizontal model grid from 2.5 to 1.25 km (SRM) or 600 to 300 m (LEM) than from 1.25 km to 600 m. This indicates a potentially stronger influence of the cloud representation on the model microphysical configuration (Sect. 3) compared to the horizontal resolution. However, the different spatiotemporal sampling of the model data might have to be considered here as well.

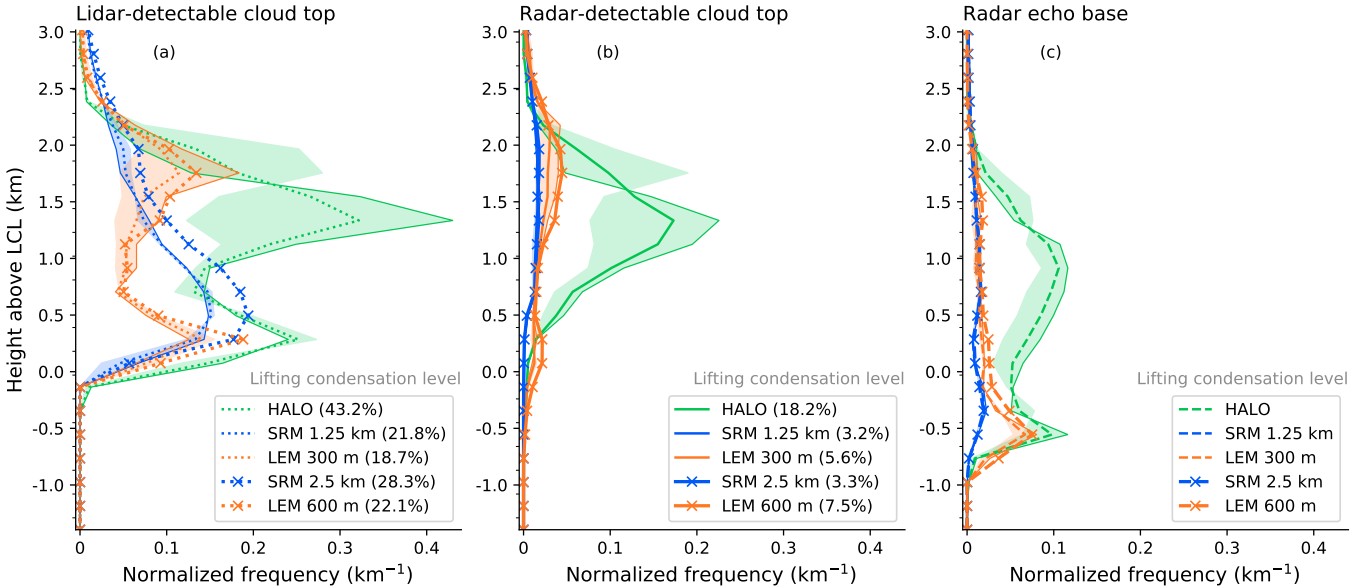

**Figure C2.** This figure extends Fig. 5 by including data from two additional domains from the SRM and LEM outputs at 2.5 km and 600 m grid spacings, respectively. For description see Fig. 5.

Figure C3 investigates the exact choice of domain and dates used in the analysis by taking subsamples from the 1.25 km SRM dataset. First the SRM 1.25 km dataset is restricted to those points that are near the LEM meteogram locations. The statistics of different cloud tops and bases in the different spatial subsets (Fig. C3) seems quite robust. Thus, we conclude, that the meteogram locations are in principle able to represent the cloud behavior of the full domain. Further, we restrict the SRM dataset to the four days for which also LEM output is available. This SRM subsample (also in Fig. C3) indicates a limited

development of a deeper stratiform of lidar-detectable cloud, which is, however, not as prominent as in the observational or LEM datasets (Fig. 5).

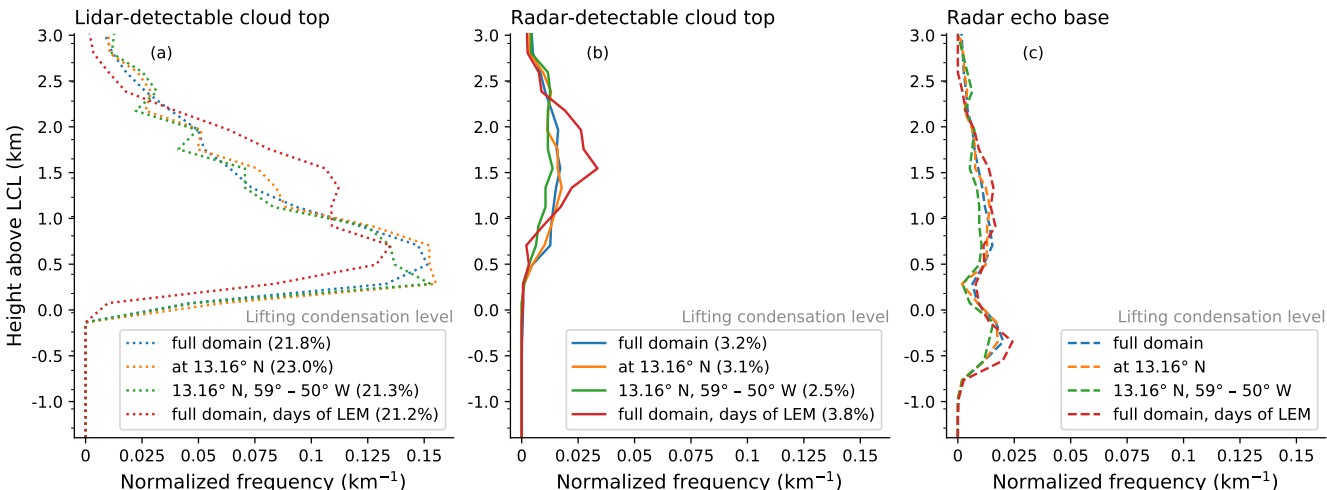

**Figure C3.** Comparison of cloud top and base heights in different spatial and temporal subsamples of the 1.25 km SRM dataset. Shown are the full domain (as in Fig. 5), data near the same latitude (13.16° N) as LEM meteogram locations, data near the same latitude and longitude range (59° – 50° W) as LEM meteogram locations, and full spatial domain but only the days with available LEM output. (a) Lidar-detectable cloud top height. (b) Radar-detectable cloud top height. (c) Radar echo base height. Total shallow 2D cloud coverage detected by lidar and radar in each dataset are given in the legends.

*Author contributions.* PK and MJ conceptualized this study. FA and SC designed the observational experiment setup and supported the interpretation of the measurements. VS supported the model interpretation. MJ performed the analysis, prepared all plots, and wrote the paper with support and input from all co-authors.

*Competing interests.* The authors declare that they have no conflict of interest.

*Acknowledgements.* The work has been supported by the German Research Foundation (Deutsche Forschungsgemeinschaft, DFG) within the DFG Priority Program (SPP 1294) "Atmospheric and Earth System Research with the Research Aircraft HALO (High Altitude and Long Range Research Aircraft)" under grant CR 111/12-1. We would like to thank Daniel Klocke and Matthias Brück for running the ICON simulations and the DKRZ for storing and supplying the data. Manuel Gutleben is thanked for making the WALES cloud top height data
available. Furthermore, we would like to acknowledge the discussion on the ICON microphysics with Axel Seifert and diagnostics with Martin Köhler and Harald Rybka from the German Weather Service (DWD). The authors gratefully acknowledge the constructive comments by two anonymous referees and by Executive Editor Astrid Kerkweg who helped improving the manuscript.

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
