# Peer review of "Multi-layer Cloud Conditions in Trade Wind Shallow Cumulus – Confronting two ICON Model Derivatives with Airborne Observations"

_Geoscientific Model Development, 2020_

## Short Comment (SC1) · 31 Mar 2020

Dear authors,

in my role as Executive editor of GMD, I would like to bring to your attention our Editorial version 1.2:

https://www.geosci-model-dev.net/12/2215/2019/

This highlights some requirements of papers published in GMD, which is also available on the GMD website in the 'Manuscript Types' section:

http://www.geoscientific-model-development.net/submission/manuscript_types.html

[Figure]

In particular, please note that for your paper, the following requirements have not been met in the Discussions paper:

- "If the model development relates to a single model then the model name and the version number must be included in the title of the paper. If the main intention of an article is to make a general (i.e. model independent) statement about the usefulness of a new development, but the usefulness is shown with the help of one specific model, the model name and version number must be stated in the title. The title could have a form such as, "Title outlining amazing generic advance: a case study with Model XXX (version Y)"."

As you are using just one model, please state in the title that you are using ICON and mention its version number.

Additionally, please provide more details on the used codes in the "Code availability" section: provide the numbers or unique identifiers of the code versions used and make sure that these versions (because this is relevant to reproduce the results of this article) are permanently archived and accessible in the future. With regard to the ICON model code I understand, that you did not perform the simulations yourself, but from your provided references (Klocke et al., 2017, and Vial et al., 2019) it is hard to find out (if at all) which model version(s) have been used and how to access the code. Please provide this information upon submission of the revised version.

Yours,

Astrid Kerkweg
* * *

---

## Referee Comment (RC1) · Anonymous Referee #1 · 21 Apr 2020

This manuscript uses simultaneous observations from a Ka-band radar and a visible/near-IR lidar aboard a high-flying aircraft to evaluate the distribution of cloud-top and cloud-base heights in shallow cumulus in two simulations with varying grid resolution. The study is focused on winter-time conditions near Barbados, site of the NARVAL experiment, where shallow (warm) clouds and precipitation dominate. After discussing how cloud tops may be determined from lidar and radar observations, and how cloud base might be identified in radar returns, the authors describe how synthetic observations are constructed from regional simulations with the Icon model at "storm-resolving" kilometer-scale horizontal resolution and "large-eddy model" at 300 meter resolution and a more complicated treatment of microphysics. The observations show

a bi-modal distribution of cloud top heights in the more sensitive lidar observations, one corresponding to well-developed shallow convection and the other attributed to very shallow clouds (as diagnosed by the small difference between the lifting condensation level and cloud-top height) which is better reproduced in the higher-resolution configuration. Sorting results by liquid water path illustrates a range of model errors with roots in some combination of microphysical and dynamical processes. The 1.25 km configuration, unsurprisingly, does not perform very well especially in thin clouds.

The manuscript is sound: all steps along the way are described in detail and appear technically correct, and the comparison between observations and model results is careful. The manuscript would be most improved by providing more context, motivation, and narrative structure, so that the inferential path readers are asked to follow becomes more clear. This might be accomplished in extensive if minor revision; the authors might also choose to undertake a more thorough re-thinking.

Readers will be especially grateful for a scientific motivation as to why it's important to look at the distributions of lidar- and radar-derived cloud top and cloud base heights. What model deficiencies does such a comparison highlight? What hypotheses might be tested by examining these statistics? Why have the authors chosen to strike this particular balance of details (e.g. with forward simulation of lidar and radar reflectivities) and abstraction (identifying cloud top and cloud base)? Additional explanation of the motivating ideas would be very welcome. The authors might also use a hypothesis to help prune away a little extraneous material.

As a related point the final section is more speculative than is satisfying. Readers will recognize that the scale of these simulations makes it difficult or impossible to produce variants. They might nonetheless expect the answer to a question or at least evidence for or against a hypothesis. The authors suggest, for example, that the inability of the storm-resolving configurations to reproduce observations in Figures 5 and 6 is due to microphysical choices or possibly vertical resolution. They neglect the possibility that the horizontal resolution is key, which one might argue based on ideas about the scale

of buoyancy production. Is it possible to distinguish between these explanations on the basis of the observations, or might the observations be interrogated differently to provide this insight?

The authors combine a sophisticated and careful calculation of synthetic radar and lidar signals, for which differences in the microphysics schemes used SRM and LEM simulations are likely germane, with a very simple analysis based on masking and stratification by liquid water path. How did the authors decide on this approach? Similarly, what motivates the assumption to ignore rain in computing lidar returns (line 200)? It's true that rain is quite unlikely to affect the lidar returns but such an assumption seems inconsistent with the level of detail used in other parts of the calculation.

The substance of the comparison lies in figures 5 and 6, which show the distribution of cloud tops detected by lidar and the tops and bases detected by radar from the observations and the two simulations. Figure 6 elaborates on Figure 5 in sorting by liquid water path. One wonders if other analyses might also be informative, especially a look at the joint distribution of lidar and radar cloud top or the joint distribution of radar top and base, depending on the questions motivating the comparison.

Section 5.3, which primarily describes Figure 6, would be enhanced by focusing on interpretation of the figure in lieu of description.

General minor points:

The referencing is quite heavily biased towards contributions from Germany. The use of sorting analyses in terms of column water vapor is due to Bretherton et al. 2005 (doi:10.1175/JAS3614.1); the sensitivity of cloud fraction to observing system has been discussed since the 1980s and covered exhaustively by Stubenrauch et al. 2013 (doi10.1175/BAMS-D-12-00117.1), etc. A more balanced and complete view would benefit readers and encourage the first author to read widely.

The manuscript's use of language would benefit from polishing by one of the senior

authors. There are more than a few typographic mistakes and a relatively large number of not-quite-standard English constructions which distract unnecessarily.

The American Meteorological Society, at least, prefers "liftING condensation level" to "liftED".

Specific minor points The sentence spanning lines 49-50 is vague, provocative, and probably unnecessary.

Section 2, describing the observations, is roughly 55 lines long. Almost half of these and one of two figures is devoted to a discussion of radar sensitivity and the definition of a threshold. Is this point important enough to warrant this level of attention?

Line 153: The line linking the parameterization suite to NWP is perhaps distracting. The authors are careful to describe differences in the microphysical approach in the SRM and LEM later, and to motivate why these might be relevant to the evaluation. This text might raise more questions than it answers.

Line 175: "forward simulations" of what?

Line 182: That forward operators need relevant state information is well-understood. Is there another point here?

Line 193: It does seem a bit odd that the synthetic lidar observations are created with a Radar Simulator while the synthetic radar observations are created with yet another package, but maybe there's nothing to be done about that.

Line 199: The relevance of water on a telescope to observations from a platform high above the clouds is not obvious.

Line 237: Is the finding that precipitation drops do not extend to the very highest reaches of shallow cumulus novel?

Line 259: A better motivation would be useful here. Readers will not expect a case study to be representative of the entire data set.

---

## Referee Comment (RC2) · Anonymous Referee #2 · 20 May 2020

Title: Multi-layer Cloud Conditions in Trade Wind Shallow Cumulus – Confronting Models with Airborne Observations

Author(s): Marek Jacob et al. MS No.: gmd-2020-14

Summary:

This study uses remote sensing observations from an airborne field campaign over the tropical Atlantic ocean upstream Barbados to assess two sets of cloud-resolving ICON simulations, one at 1.25 km grid spacing (SRM) with one-moment microphysics and the other at 300 m grid spacing (LEM) with two-moment microphysics. The model-observation comparison is based on forward-simulated model output to mimic what the

aircraft radar and lidar would see given the atmospheric state and the microphysical assumptions in the model. The authors find the LEM to reproduce the observed bi-modal cloud top height distribution seen by the lidar, while the SRM fails to represent the upper mode. Stratifying the results into different LWP classes shows that also the LEM model has significant deficiencies in its representation of the radar- and lidar-detected cloud top and base height distributions.

This is a nice study that fits well into the scope of GMD. The use of the forward simula-tion gives interesting new insights about the deficiencies of cloud-resolving simulations in representing shallow cumulus clouds. My main comments regard a more thorough comparison of the representativeness of the selected LEM and SRM profiles, and an analysis of the uncertainty of the forward-simulation and the sensitivity of the results to the microphysical model assumptions.

My general comments are detailed in the following, as well as more specific comments and typographical suggestions.

GENERAL COMMENTS:

1. Comparability of selected LEM and SRM profiles:

Vial et al. 2019 showed in their Figure 5 that the 1.25km SRM has a larger cloud cover than the 300-m LEM, especially due to larger contributions from clouds with cloud tops > 1.3km. So I'm surprised that your results here are so different. This might be due to the different microphysical assumptions, but could also be due to the different domains and days used for the SRM vs. the LEM.

For the LEM, it seems that you are using data from only 10 grid points on 4 days, all sampled at the same latitude. Due to the high temporal resolution of the meteogram output this may give you a lot of profiles, but they will all be highly (auto)correlated. The LEM thus samples much less variable conditions than the SRM. To allow for a more robust and fair comparison of the LEM and the SRM, a comparison of the cloud

fractions and/or cloud top height distributions of the LEM and SRM for the same domain and the same days should be made. This would establish how representative the meteogram data is.

As the necessary input for the forward-simulator is available only for the meteogram points of the LEM, it might be difficult to use the model output of the full LEM domain to do the forward simulation. You could just use the mean and variability of the parameters from the meteogram points to constrain the forward-simulator, which can then be applied to the entire domain.

Additionally, to understand how much of the forward-simulated SRM-LEM differences come from the different microphysics, I find it important to first show a comparison of the cloud-top height distributions of the two models without using the forward-simulator. This should also be compared to a best-guess observational cloud-top height distribution, either from the lidar alone or from a combination of the lidar and radar-detected clouds. For the lidar, you mention that clouds with liquid water content exceeding 10ˆ-7 kg/kg are detected. So you could apply this same threshold to the LEM and SRM simulations (for the SRM, also the sub-grid cloudiness will have to be taken into account).

Apart from showing the frequency distribution as in Figure 5, it might also be worth comparing the cumulative distributions as e.g. in Medeiros et al. 2010 (Figure 7) or van Zanten et al. 2011 (also Figure 7), with the lowermost level representing the total cloud cover.

Other questions regarding the simulations are:

- Are there any spin-up issues at the beginning of the simulations and is it feasible to use the LEM simulations already from 12 UTC on, i.e. just 3h after initialization? - In the appendix you mention that the SRM uses a diagnostic cloud s cheme in addition to the prognostic cloud scheme. This is an important detail that should already be mentioned in section 3. Furthermore, could you describe how the 'prognostic' cloud scheme works? Is it just a simple saturation adjustment?

- Difference in vertical resolution: In L420 you mention this as a potential reason for the underrepresentation of inversion cloud in the SRM. I guess the 1.25km SRM version that was used to drive the LEMs should have 150 levels – so in case this model output was saved, you could use this SRM version to verify whether the vertical resolution is indeed the reason for the reduced anvils. Otherwise you could try to better understand the influence of the horizontal resolution on the anvil cloud amount by comparing the 300m-LEM to the next coarser LEM nest.

2. Uncertainty and sensitivity of forward-simulations to model assumptions

I'm not very familiar with forward simulators, but I feel that it would be important to analyze and discuss the uncertainty of the forward simulations, and how this might influence the results. You mention that the forward simulator has to be configured such that the PSD used in the forward-simulator matches the PSD of the model as good as possible. I assume that there is some uncertainty involved in this process, and it would be good to show or discuss this more explicitly.

I would also appreciate if you could show somewhere what the variability of the input fields for the scale parameters are in the LEM, i.e. how variable the number concentrations are. This information can also help constrain a forward-simulation using the entire model domain of the LEM.

Also, I think that you could learn more about the potential deficiencies in the model microphysics by playing around with the forward simulator and feeding it with slightly adjusted input microphysics parameters. What would have to be different in the microphysics to render the simulations more comparable to the observations, given the simulated mass mixing ratios? You could try to understand how a slight change of the fixed parameters of the SRM one-moment scheme would influence the radar-detectable cloud fraction. Given that the droplet radius is so important for the radar-detectability, Figure 3c might look very different if you'd just fed the forward-simulator with slightly different number concentration parameters. For the LEM, You could also

prescribe the mean number concentrations of the LEM as fixed parameter to mimic what a one-moment microphysics scheme would do.

A more thorough analysis of the uncertainty and sensitivity of the forward-simulations would render the manuscript scientifically more interesting, and should allow you to make your discussion in Section 6 more robust and less speculative.

SPECIFIC COMMENTS:

- Definition of cloud modes / types (e.g. L263-275): Please better define what you mean with 'thermal driven' mode resp. 'shallow convection' mode. You could also use well-established classifications or definitions such as the 'forced, active and passive' categories of Stull 1985, or the definitions from the cloud atlas of the World Meteorological Organization that were used in Vial et al. 2019 JAMES.

- Cloud-top height detection: I think it is never explicitly written whether you only consider the first-detected highest cloud-top height, or whether you also consider 2nd or pot. 3rd cloud-top heights in case of multilayered cloud scenes. Please mention this explicitly.

- Referencing previous literature:

o The bi-modal distribution of trade cumuli in the vicinity of Barbados has been extensively studied by Nuijens et al. using data from the Barbados Cloud Observatory. Please refer to Nuijens et al. 2014 QJRMS in L56 and also later on in the manuscript. (Related to this, in L262 it would be good to mention that the 30% dominance of the upper mode vs. the lower mode is opposite when considering ground-based observations (see Nuijens et al. 2014).)

o Observations from the CSET field campaign in the eastern pacific presented in O et al. 2018 GRL and Wood et al. 2018 JAS revealed the common occurrence of persistent thin outflow layers with very low droplet concentrations (the authors refer to 'veil' clouds and ultra-clean layers). It may be good to cite and discuss these papers in the context

of the present results.

o The referencing for the first two sentences in the introduction should be improved.

- The LCL computation from the dropsondes: Can you say how many dropsondes are used to interpolate the LCL? And by how much they are separated in space and time on average?

- Differences between the western and eastern part of the domain related to cloud deepening: Not only is there a difference in the height of the upper mode, but also in the normalized frequencies of the upper mode, with the deeper western half having a reduced frequency compared to the shallower eastern half. This, and also the insensitivity of the lower mode, was also shown in LES of Vogel et al. 2020 QJRMS.

- Section 5.3: The discussion of the results in this section should be better structured and more focused on the most important features. It is not always clear what is compared to what, and there is a lot of switching around between LWP categories, the observations and the different models. I also spotted a lot of typographical errors that should be corrected (e.g. L347 partial coverage; L363 that such a cloud doesn't need any contribution...)

- Figure 1: This figure could be improved. Please zoom more into the area of flight operations (only showing e.g. 7°N to 20°N), make sure that all flight paths are visible and not overlapping, and add markers/crosses for the dropsonde locations.

- Figure 5: What exactly does the cloud fraction in the legend refer to? Is it just the maximum cloud fraction? It would be nice to give the total projected cloud cover instead of a cloud fraction, as this would give a sense of the total cloudiness.

- Figure A1: similar to the above, in the caption you mix cloud cover and cloud fraction, but I guess you mean the same thing.

TYPOGRAPHICAL SUGGESTIONS:

- L65 (and everywhere else): model data –> model output

- L89: maybe add 'first phase of the NARVAL... Often referred to as NARVAL1 in other studies.

- L97: this sentence is a bit odd there and should be moved down after L104, maybe adj. it to 'The following subsections describe...'

- L98: form –> from

- L99: better than 20 g/m-2 and 10% (?)

- L107: radar –> reflectivity

- L116: Ragged point –> Deebles point

- L124: a clear frequency maximum?

- L206 & other instances throughout the manuscript: remove commas before 'that' → "it has to be noted that..."

- L212: by –> be

- L255-257: could be omitted.

- L285: what do you mean with shallow clouds here? (Please also see my specific comment on the definition of the cloud modes above)

- L306: remove 'is'

- L316: simulated infrequently –> underrepresented

- L321ff: The thresholds for the LWP classes are different from Figure 6 (i.e. lower bounds not given)

- L338: Ref to figure 5 and not figure 4

- L383: liar –> lidar

- L419: a gap of what? Please be more specific

- L424: might be an...

- L491: remove 'with'

---

## Author Comment (AC1) · 5 Sep 2020

**Authors' Reply**

The authors would like to thank the referees and the Executive Editor for their constructive feedback, that helped in improving the manuscript. In the following, all revision comments are addressed and the resulting edits are included in the following way: The comments are repeated and responses are given below. Changes made in the manuscript are indicated in blue. Figure numbers with "R" correspond to figures in this reply. In the attached manuscript, red and blue indicate removed and added text, respectively.

**Reply to Anonymous Referee #1**

**Comment:** This manuscript uses simultaneous observations from a Ka-band radar and a visible/near-IR lidar aboard a high-flying aircraft to evaluate the distribution of cloud- top and cloud-base heights in shallow cumulus in two simulations with varying grid resolution. The study is focused on winter-time conditions near Barbados, site of the NARVAL experiment, where shallow (warm) clouds and precipitation dominate. After discussing how cloud tops may be determined from lidar and radar observations, and how cloud base might be identified in radar returns, the authors describe how synthetic observations are constructed from regional simulations with the Icon model at "storm- resolving" kilometer-scale horizontal resolution and "large-eddy model" at 300 meter resolution and a more complicated treatment of microphysics. The observations show a bi-modal distribution of cloud top heights in the more sensitive lidar observations, one corresponding to well-developed shallow convection and the other attributed to very shallow clouds (as diagnosed by the small difference between the lifting conden-sation level and cloud-top height) which is better reproduced in the higher-resolution configuration. Sorting results by liquid water path illustrates a range of model errors with roots in some combination of microphysical and dynamical processes. The 1.25 km configuration, unsurprisingly, does not perform very well especially in thin clouds.

The manuscript is sound: all steps along the way are described in detail and appear technically correct, and the compari-son between observations and model results is careful. The manuscript would be most improved by providing more context, motivation, and narrative structure, so that the inferential path readers are asked to follow becomes more clear. This might be accomplished in extensive if minor revision; the authors might also choose to undertake a more thorough re-thinking.

**Response:** We hope to satisfy this comment through modifications and responses to your individual comments as outlined below and also following Referee #2.

**Comment:** Readers will be especially grateful for a scientific motivation as to why it's important to look at the distributions of lidar- and radar-derived cloud top and cloud base heights. What model deficiencies does such a comparison highlight? What hypotheses might be tested by examining these statistics? Why have the authors chosen to strike this particular balance of details (e.g. with forward simulation of lidar and radar reflectivities) and abstraction (identifying cloud top and cloud base)? Additional explanation of the motivating ideas would be very welcome. The authors might also use a hypothesis to help prune away a little extraneous material.

**Response:** We realized, that a better motivation could help the reader. Thus we extend the introduction and explain: "To assess vertically resolved cloudiness and shallow convection, we compare the vertical cloud boundaries. The exact location of the cloud boundaries is the major parameter determining the heating rate profile. Further, the cloud top height is an indicator of the convective activity and therefore allows assessing the model physics with observations indirectly. While cloud fraction at cloud base is rather robust among model assumptions, cloud fraction near the trade inversion varies strongly (Vogel et al., 2020). Clouds near the inversion are often very thin (O et al., 2018) and therefore the problem of instrument sensitivity (e.g., Stubenrauch et al., 2013) can provide different answers about their exact vertical placement."

**Comment:** As a related point the final section is more speculative than is satisfying. Readers will recognize that the scale of these simulations makes it difficult or impossible to produce variants. They might nonetheless expect the answer to a question or at least evidence for or against a hypothesis. The authors suggest, for example, that the inability of the storm-resolving con-figurations to reproduce observations in Figures 5 and 6 is due to microphysical choices or possibly vertical resolution. They

neglect the possibility that the horizontal resolution is key, which one might argue based on ideas about the scale of buoyancy production. Is it possible to distinguish between these explanations on the basis of the observations, or might the observations be interrogated differently to provide this insight?

r50 **Response:** We investigated the effect of horizontal resolution by looking at another, coarser domain from the SRM and LEM outputs. They are the SRM at 2.5 km and LEM at 600 m grid spacing. The differences between the two SRMs or two LEMs are significantly smaller than those between the 1.25 km SRM and 600 m SRM (see Fig. R1). However, the different spatiotemporal sampling of the model data might have to be considered here as well. We make this more clear in the manuscript, provide Fig. R1 in the new appendix C, and consider the influence of horizontal resolution in the conclusions as well.

[Figure]

**Figure R1.** This figure extends Fig. 5 by including data from two additional domains from the SRM and LEM outputs at 2.5 km and 0.6 km grid spacings, respectively. For description see Fig. 5.

r55 **Comment:** The authors combine a sophisticated and careful calculation of synthetic radar and lidar signals, for which differences in the microphysics schemes used SRM and LEM simulations are likely germane, with a very simple analysis based on masking and stratification by liquid water path. How did the authors decide on this approach?

**Response:** The approach was chosen based on the available observational dataset. As mentioned in the reply to the second comment, we wanted to emphasize the necessity to account for different sensitivities of cloud observations. As the lidar signal is often attenuated shortly below cloud top, we choose to constrain our analysis to the height of the lidar-detectable cloud top.

r60 To then unify the analysis of the radar data with the lidar data, similar masking was applied to the radar data considering that the radar signal is fully attenuated only rarely. This is reflected in introduction of the manuscript (ll 85– 90 in the version below)

r65 **Comment:** Similarly, what motivates the assumption to ignore rain in computing lidar returns (line 200)? It's true that rain is quite unlikely to affect the lidar returns but such an assumption seems inconsistent with the level of detail used in other parts of the calculation.

**Response:** The cloud top height estimation is not expected to change from the raindrop scattering based on the assumption of an optically thin raindrop class. However, we reconfigured CR-SIM to also consider raindrops in the lidar calculation. The related cloud top statistics are shown in Fig. R2 and confirm our assumption, i.e., the rain does not affect the lidar forward

r70 simulation. We clarified the sentence in the revised manuscript as follows. "As raindrops near cloud top are optically thin,

[Figure]

**Figure R2.** Lidar-detectable cloud top hight distributions from the LEM dataset testing the sensitivity of the lidar forward simulation towards raindrops. Thick are from forward simulations considering cloud droplets only (Like Figures 5a and 7a, d, g, j, and m). Thin lines with x markers are from forward simulations considering cloud droplets and raindrops. Colors denote different LWP regimes.

if present in the models at all, we decided to simplify the forward simulation of the backscatter lidar and thus ignore their contributions."

**Comment:** The substance of the comparison lies in figures 5 and 6, which show the distribution of cloud tops detected by lidar and the tops and bases detected by radar from the observations and the two simulations. Figure 6 elaborates on Figure 5 in sorting by liquid water path. One wonders if other analyses might also be informative, especially a look at the joint distribution of lidar and radar cloud top or the joint distribution of radar top and base, depending on the questions motivating the comparison.

**Response:** Figure R3 shows respective joint distributions. The direct comparison of lidar and radar cloud top and base heights provides an interesting view on the data, even though it limits the analysis to the cases that provide a radar signal. The clouds observed from HALO are often either about 200 to 400 m thin (along the diagonal) and at about 1 km above the LCL or they precipitate (on the left) with similar cloud top heights. The SRM dataset shows a similar mode of thin clouds at generally lower heights, and precipitating clouds with deeper cloud tops compared to the observations. In the LEM, the non precipitating clouds develop with a wider range of depths. The precipitating clouds however, have a similar top height like in the SRM. Furthermore, there are a couple of observed cases (right of the diagonal in Figures 6 d-f) in which the lidar-detected cloud top heights are below the radar base heights. These signals relate to lateral raindrop transport out of the precipitating core with a patch of cloud beneath. Such cases also occur in the LEM dataset and less frequently in the SRM dataset. The smaller grid spacing in the LEM could be responsible the higher likelihood for lateral transport of raindrops into a neighbor grid cell in the LEM compared to the SRM.

We included Fig. R3 as Fig. 6 and this discussion into the manuscript.

[Figure]

**Figure R3.** Relation of radar-detectable cloud top height (a-c) and lidar-detectable cloud top height (d-f) above LCL versus radar echo base height above LCL. Based on HALO observations (a, d), 1.25 km SRM dataset (b, e), and 300 m LEM dataset (c, f).

**Comment:** Section 5.3, which primarily describes Figure 6, would be enhanced by focusing on interpretation of the figure in lieu of description.

r95 **Response:** We agree with the reviewer and reorganized section 5.3. Furthermore, we added an additional figure (Fig. 8) which summarizes the key cloud features as a function of LWP. The figure shows the relation of key cloud features as a function of LWP more easily. This figure also shows the influence of model configuration (SRM or LEM) versus horizontal resolution by including the outputs of the SRM and LEM at two additional resolutions and supports the discussion.

**General minor points:**

r100 **Comment:** The referencing is quite heavily biased towards contributions from Germany. The use of sorting analyses in terms of column water vapor is due to Bretherton et al. 2005 (doi:10.1175/JAS3614.1); the sensitivity of cloud fraction to observing system has been discussed since the 1980s and covered exhaustively by Stubenrauch et al. 2013 (doi10.1175/BAMS-D-12-00117.1), etc. A more balanced and complete view would benefit readers and encourage the first author to read widely.

**Response:** The reviewer is right: We performed a more detailed literature review and included additional references. Further, we removed the sentence related to water vapor sorting from the manuscript as it provided superfluous information, which could be misinterpreted.

**Comment:** The manuscript's use of language would benefit from polishing by one of the senior authors. There are more than a few typographic mistakes and a relatively large number of not-quite-standard English constructions which distract unnecessarily.

**Response:** We performed a more thorough proof reading and revised typos also considering the comments from referee #2.

**Comment:** The American Meteorological Society, at least, prefers "liftING condensation level" to "liftED".

**Response:** We adjusted the manuscript to the AMS recommendations and updated the text and all figures accordingly.

**Comment:** Specific minor points
The sentence spanning lines 49-50 is vague, provocative, and probably unnecessary.

**Response:** We changed the related sentence to "A reason to initiate the NARVAL expeditions was to extend satellite observations. This can test which cloud variables are sufficiently resolved from space and which characteristics benefit from higher spatial resolution in respect to shallow cumulus clouds." and hope this motivation is clearer and less provocative.

**Comment:** Section 2, describing the observations, is roughly 55 lines long. Almost half of these and one of two figures is devoted to a discussion of radar sensitivity and the definition of a threshold. Is this point important enough to warrant this level of attention?

**Response:** We realize, that this discussion might distract the reader. Because the radar sensitivity aspect is important we moved the details and derivation of the threshold into an appendix and only kept the main message in section 2.1).

**Comment:** Line 153: The line linking the parameterization suite to NWP is perhaps distracting. The authors are careful to describe differences in the microphysical approach in the SRM and LEM later, and to motivate why these might be relevant to the evaluation. This text might raise more questions than it answers.

**Response:** We removed the distracting and irrelevant mention of NWP and shortened the sentence to "The SRM is run without a convection parameterization." To highlight an important difference of the LEM, we add a link to the applied Smagorinsky scheme for turbulence.

**Comment:** Line 175: "forward simulations" of what?

**Response:** We specified the section title and changed it to "Radar and Lidar Forward Simulations".

**Comment:** Line 182: That forward operators need relevant state information is well-understood. Is there another point here?

**Response:** In addition to the numerical output fields of the model, assumptions of the model like the drop size distributions have to be considered in the forward simulation. To clarify this, we added the explanation "This means that the same PSD shape and parameters have to be used in the simulator as assumed in the model".

**Comment:** Line 193: It does seem a bit odd that the synthetic lidar observations are created with a Radar Simulator while the synthetic radar observations are created with yet another package, but maybe there's nothing to be done about that.

**Response:** We added a clarification that the lidar backscatter was "simulated using the lidar simulation capabilities of the Cloud Resolving Model Radar Simulator".

**Comment:** Line 199: The relevance of water on a telescope to observations from a platform high above the clouds is not obvious.

**Response:** We clarified the relevance of this statement by appending the sentence "Furthermore, as the airborne lidar is not affected by liquid collection on the telescope during raining conditions, there is no need to account for such effect" by "as it

would be for a ground based lidar".

**Comment:** Line 237: Is the finding that precipitation drops do not extend to the very highest reaches of shallow cumulus r155 novel?
**Response:** True, this is unsurprisingly but different from the forward simulation as pointed out later in the manuscript. We noted this observational finding to relate it to the forward simulations at a later point of the manuscript.

**Comment:** Line 259: A better motivation would be useful here. Readers will not expect a case study to be representative of r160 the entire data set.
**Response:** We changed the introduction of the subsection to "After introducing and discussing the approach in a case study, all observations and simulations are jointly analyzed in this section.".

**Reply to Anonymous Referee #2**

r165 **Comment:** Summary: This study uses remote sensing observations from an airborne field campaign over the tropical Atlantic ocean upstream Barbados to assess two sets of cloud-resolving ICON simulations, one at 1.25 km grid spacing (SRM) with one-moment microphysics and the other at 300 m grid spacing (LEM) with two-moment microphysics. The model– observation comparison is based on forward-simulated model output to mimic what the aircraft radar and lidar would see given the atmospheric state and the microphysical assumptions in the model. The authors find the LEM to reproduce the observed bi-r170 modal cloud top height distribution seen by the lidar, while the SRM fails to represent the upper mode. Stratifying the results into different LWP classes shows that also the LEM model has significant deficiencies in its representation of the radar- and lidar- detected cloud top and base height distributions. This is a nice study that fits well into the scope of GMD. The use of the forward simulation gives interesting new insights about the deficiencies of cloud-resolving simulations in representing shallow cumulus clouds. My main comments regard a more thorough comparison of the representativeness of the selected LEM and r175 SRM profiles, and an analysis of the uncertainty of the forward-simulation and the sensitivity of the results to the microphysical model assumptions. My general comments are detailed in the following, as well as more specific comments and typographical suggestions.

**GENERAL COMMENTS:**

**Comment:** 1. Comparability of selected LEM and SRM profiles:
r180 Vial et al. 2019 showed in their Figure 5 that the 1.25km SRM has a larger cloud cover than the 300-m LEM, especially due to larger contributions from clouds with cloud tops > 1.3km. So I'm surprised that your results here are so different. This might be due to the different microphysical assumptions, but could also be due to the different domains and days used for the SRM vs. the LEM.
**Response:** In our Fig. 5 (Fig. 5 of the initially submitted manuscript) the 1.25-km SRM has a total projected 2D cloud coverage r185 of 21.8 %, which is slightly higher than the cloud covered of the 300-m LEM (18.7 %). These numbers are slightly different from the ones presented in Fig. 5 by Vial et al. (2019). Vial et al. denote the total cloud covered as 28 and 20 % for the 1.25-km SRM and 300-m LEM, respectively. This means, their difference is larger, which is likely due to the reduced sample (domain, days) and the fact that we are analyzing day-time data only.
To better quantify these sampling effects, we checked the exact choice of domain and dates used in our analysis by selecting r190 subsets from the SRM dataset. To clarify this in the manuscript we added Fig. C3 into appendix C and explain the sampling effect as follows. " Figure C3 investigates the exact choice of domain and dates used in the analysis by taking subsamples from the 1.25 km SRM dataset. First the SRM 1.25 km dataset is restricted to those points that are near the LEM meteogram locations. The statistics of different cloud tops and bases in the different spatial subsets (Fig. C3) seems quite robust. Thus, we conclude, that the meteogram locations are in principle able to represent the cloud behavior of the full domain. Further, r195 we restrict the SRM dataset to the four days for which also LEM output is available. This SRM subsample (also in Fig. C3) indicates a limited development of a deeper stratiform of lidar-detectable cloud, which is, however, not as prominent as in the

noneobservational or LEM datasets (Fig. 5)." Thus the conclusions from our original analysis hold.

**Comment:** For the LEM, it seems that you are using data from only 10 grid points on 4 days, all sampled at the same latitude. Due to the high temporal resolution of the meteogram output this may give you a lot of profiles, but they will all be highly (auto)correlated. The LEM thus samples much less variable conditions than the SRM. To allow for a more robust and fair comparison of the LEM and the SRM, a comparison of the cloud fractions and/or cloud top height distributions of the LEM and SRM for the same domain and the same days should be made. This would establish how representative the meteogram data is.

**Response:** See previous response.

**Comment:** As the necessary input for the forward-simulator is available only for the meteogram points of the LEM, it might be difficult to use the model output of the full LEM domain to do the forward simulation. You could just use the mean and variability of the parameters from the meteogram points to constrain the forward-simulator, which can then be applied to the entire domain.

**Response:** Unfortunately, no 3D output of rain water content from the LEM is available, such that the analysis of radar-detectable cloud and precipitation and not be extended to the full LEM domain. However, we hope that the representativity analysis in Fig. C3 can answer this question sufficiently.

**Comment:** Additionally, to understand how much of the forward-simulated SRM-LEM differences come from the different microphysics, I find it important to first show a comparison of the cloud-top height distributions of the two models without using the forward-simulator. This should also be compared to a best-guess observational cloud-top height distribution, either from the lidar alone or from a combination of the lidar and radar-detected clouds. For the lidar, you mention that clouds with liquid water content exceeding $10^{-7}$ kg/kg are detected. So you could apply this same threshold to the LEM and SRM simulations (for the SRM, also the sub-grid cloudiness will have to be taken into account).

**Response:**
Forward operators are important to translate sensors threshold into model variables. For example, in the case of one-moment lidar forward simulations, one can read from Fig. 2a that the observational threshold of BSR > 20 corresponds to a model threshold of $q_c > 10^7$ to $10^{-6}$ kg kg$^{-1}$. As most of the grid cells show higher lidar backscatter ratio values (color shading in Fig. 2a), such $q_c$ threshold identifies almost all grid cells which would be also lidar-detectable as derived from the forward simulator setup. Likewise radar detection rules can be estimated for the one-moment microphysics from Fig. 2c suggesting the signal detection criterion: $q_r > 10^{-7}$ kg kg$^{-1}$ (upper left branch of the distribution) or $q_c > 8 \times 10^{-4}$ kg kg$^{-1}$ (lower left).

As expected, the lidar cloud layer detection from the forward simulators can be approximated with the given thresholds to a very high degree of agreement as shown in Fig. R4a. In case of the one-moment SRM microphysics, the threshold-based radar approximation also provides good estimates of the full forward simulation (Figures R4 b and c), once appropriate thresholds are found using the forward simulator. In contrast, the more complex two moment microphysical scheme of the LEM shows that simple thresholds on the model hydrometeor water contents are not sufficient to represent the cloud statistics one could get from more advanced forward simulators. As this analysis shows, there is no unique threshold for cloud definition. Using a threshold for the water load is as arbitrary as any other definition. However, the forward operator approach ensures a fair comparison between observations and models in the same quantity space. To not confuse the reader, we would like to avoid a further excursion from the initial study by discussing these water load thresholds in the manuscript.

Regarding the sub-grid cloudiness, we now explain: "An appropriate analysis of the cloud top height distribution of the sub-grid cloudiness in the SRM would require an assumption on the horizontal overlap of sub-grid clouds within a model column. To circumvent an assumption on this, the influence of the sub-grid clouds is analyzed in terms of the CF profile." This analysis is already covered in appendix B, but we added the motivation above to clarify why appendix B uses a slightly different metric than the main part.

[Figure]

**Figure R4.** Comparison of cloud top and base heights derived from forward simulations and direct thresholds on model output variables. Thick lines denote distributions from forward simulations, thin lines with x marker denote distributions from direct thresholds. (a) Lidar-detectable cloud top height. (b) Radar-detectable cloud top height. (c) Radar echo base height.

**Comment:** Apart from showing the frequency distribution as in Figure 5, it might also be worth comparing the cumulative distributions as e.g. in Medeiros et al. 2010 (Figure 7) or van Zanten et al. 2011 (also Figure 7), with the lowermost level representing the total cloud cover.

**Response:** We thank the reviewer for the good idea and added the corresponding plots (Fig. C1 in the appendix) which nicely summarize the results.

**Comment:** Other questions regarding the simulations are:
- Are there any spin-up issues at the beginning of the simulations and is it feasible to use the LEM simulations already from 12 UTC on, i.e. just 3h after initialization? - In the appendix you mention that the SRM uses a diagnostic cloud scheme in addition to the prognostic cloud scheme. This is an important detail that should already be mentioned in section 3. Furthermore, could you describe how the 'prognostic' cloud scheme works? Is it just a simple saturation adjustment?

**Response:** The SRM was initialized at 00 UTC on each day. We use data starting from 12 UTC after granting the model 12 hr for spin-up. The LEM was initialized at 09 UTC using outputs from the SRM. The coarsest LEM simulation has the same resolution as the SRM such that nested LEM should be usable after a shorter spin-up time. Thus, we chose to grant the LEM 3 hr of spin-up. We added details about the spin-up time to the manuscript.
Regarding the prognostic scheme we add that "It is assumed, that turbulent perturbations distribute the total water content in a probability density function (PDF) of rectangular shape centered around its prognostic grid-box mean. The supersaturated part of this PDF is then interpreted as diagnostic cloud cover and $q_{c,dia}$" in the appendix B. Further, we mention the diagnostic scheme in Sect. 3 now.

**Comment:** - Difference in vertical resolution: In L420 you mention this as a potential reason for the underrepresentation of inversion cloud in the SRM. I guess the 1.25km SRM version that was used to drive the LEMs should have 150 levels – so in case this model output was saved, you could use this SRM version to verify whether the vertical resolution is indeed the reason for the reduced anvils. Otherwise you could try to better understand the influence of the horizontal resolution on the anvil cloud amount by comparing the 300m-LEM to the next coarser LEM nest.

**Response:** The LEM was directly forced with the 75-level SRM. Thus, we cannot analyze this point directly. However, we include data from the next coarser LEM nest with 600 m grid spacing in the analysis. The statistic of that dataset is much more similar to the 300-m LEM, than to the 1250-m SRM. This can also be seen in the new Figures 8 and C2 in the appendix as also

r270 motivated in third comment by referee #1.

**Comment:** 2. Uncertainty and sensitivity of forward-simulations to model assumptions

I'm not very familiar with forward simulators, but I feel that it would be important to analyze and discuss the uncertainty of the forward simulations, and how this might influence the results. You mention that the forward simulator has to be configured

r275 such that the PSD used in the forward-simulator matches the PSD of the model as good as possible. I assume that there is some uncertainty involved in this process, and it would be good to show or discuss this more explicitly.

**Response:** This process involves very little uncertainty, as it is about implementing a PSD which matches the one assumed by the model in the forward simulator. To make this more clear and dispel doubts we removed the addendum "as possible" from the manuscript. The related sentence is now: "The forward simulator has to be configured such that the PSD used to simulate

r280 hydrometeor characteristics matches the PSD assumed in the atmospheric model accurately."

**Comment:** I would also appreciate if you could show somewhere what the variability of the input fields for the scale parameters are in the LEM, i.e. how variable the number concentrations are. This information can also help constrain a forward-simulation using the entire model domain of the LEM.

r285 **Response:** In order to better illustrate the variability we added the quartiles of $N_c$ and a plot for the $q_r - N_r$ relation from the LEM output. Further, we extended the discussion of that figure and the sensitivity of forward-simulation. The last paragraph of Sect. 4 related to the LEM is now: "The median of $N_c$ is $3.8 \times 10^8 \, \mathrm{kg}^{-1}$ with an interquartile range between 2.0 an $6.8 \times 10^8 \, \mathrm{kg}^{-1}$ for grid cells containing any cloud water. This means $N_c$ is mostly larger in the LEM than in the SRM (fixed to $N_c = 2 \times 10^8 \, \mathrm{kg}^{-1}$, Tab. 1), and that cloud droplets are smaller for the same $q_c$ in the LEM." ... "Different to cloud water,

r290 the radar reflectivity of rain is in many cases amplified in the two-moment scheme compared to the one-moment simulation, such that also some grid cells with lower $q_r$ are above the radar detection threshold. This indicates in general larger and fewer raindrops in the LEM than in the SRM for the same $q_r$ as also depicted in Fig. 3."

**Comment:** Also, I think that you could learn more about the potential deficiencies in the model microphysics by playing

r295 around with the forward simulator and feeding it with slightly adjusted input microphysics parameters. What would have to be different in the microphysics to render the simulations more comparable to the observations, given the simulated mass mixing ratios? You could try to understand how a slight change of the fixed parameters of the SRM one-moment scheme would influence the radar-detectable cloud fraction. Given that the droplet radius is so important for the radar-detectability, Figure 3c might look very different if you'd just fed the forward-simulator with slightly different number concentration parameters.

r300 For the LEM, You could also prescribe the mean number concentrations of the LEM as fixed parameter to mimic what a one-moment microphysics scheme would do.

**Response:** We concur with the reviewer that instrument simulators generate "synthetic" measurements that we can compare against real observations, thus, offers an alternative methodology for "adjusting" model microphysics. In this framework, observations can be a powerful constraint, however, at the same time, the model should contain enough information (i.e. mir305 crophysical moments) to facilitate the "adjustment".

To better explain the relationship between microphysical quantities, we added to the manuscript that "For a fixed water mixing ratio, a change by a factor of $\alpha$ in the number concentration ($N_c' = \alpha \, N_c$) will result to a change in the radar reflectivity in dBZ from $Z$ to $Z' = Z - 10 \log_{10} \alpha$. Thus, if we double the $N_c$ ($\alpha = 2$), the $Z$ will reduce by 3 dB. By the same token, if we change the hydrometeor diameter by a factor of $\alpha$ ($D' = \alpha \, D$), then the radar reflectivity will be $Z' = Z + 30 \log_{10} \alpha$".

r310 Thus, as the reviewer suggest, if we attempt to match the observed radar reflectivity with the simulator output, the preference will be to adjust the size of the droplets if the difference is large. If the difference is small (1–2 dB), we can not use the radar reflectivity to adjust this properly in the model since both parameters (N and D) can address small differences. In addition to the direct impact on radar reflectivity, a change in droplet number or size would also influence the dynamics of the model as $N$ is a prognostic variable. This could change the overall model cloud statistics significantly, which would be an interesting topic

r315 for further research.

**Comment:** A more thorough analysis of the uncertainty and sensitivity of the forward-simulations would render the manuscript scientifically more interesting, and should allow you to make your discussion in Section 6 more robust and less speculative.

**Response:** We realize that many readers are not so familiar with forward operators and therefore added some of the discussion of the above points into the manuscript.

**Comment:** SPECIFIC COMMENTS:
- Definition of cloud modes / types (e.g. L263-275): Please better define what you mean with 'thermal driven' mode resp. 'shallow convection' mode. You could also use well-established classifications or definitions such as the 'forced, active and passive' categories of Stull 1985, or the definitions from the cloud atlas of the World Meteorological Organization that were used in Vial et al. 2019 JAMES.

**Response:** We appreciate this suggestion and moved our previous definition from Sect. 5.3 to the introduction of Sect. 5. There we added the following. "To ease the following discussion, we define three layers in which the lidar and radar signals occur. Every signal below LCL is in the 'precipitation' layer. Typically, only the radar base is in this layer. Clouds with their tops within 600 m above LCL are called 'very shallow clouds' following the definitions by Vial et al. (2019). Vial et al. defined this mode in terms of an absolute top height below 1.3 km which corresponds to a similar height considering that the LCLs in the dropsonde, SRM, and LEM datasets in this study have typical heights of $720 \pm 135$, $763 \pm 144$, and $777 \pm 121$ m, respectively. Cumulus humilis is a typical representative of these very shallow clouds but in principle this class contains also small parts of deeper but slanted clouds. More active clouds can grow deeper than these very shallow clouds until they encounter the trade inversion and are forced to form a lateral outflow which is often perceivable as a stratiform layer. Stratiform remnants of such shallow convection can last for hours and thus much longer than the original convective core (Wood et al., 2018). We summarize all cloud signals above LCL + 600 m as 'stratiform' mode, acknowledging also contributions from active cores."
We adjust the subsequent discussion in the manuscript accordingly. By following the definitions by Vial et al. (2019), we reduced the vertical extend from the lower cloud mode from LCL + 1000 m to LCL + 600 m and thus adjusted cloud fractions in corresponding statements.

**Comment:** - Cloud-top height detection: I think it is never explicitly written whether you only consider the first-detected highest cloud-top height, or whether you also consider 2nd or pot. 3rd cloud-top heights in case of multilayered cloud scenes. Please mention this explicitly.

**Response:** We clarified that only the uppermost cloud tops and lowest bases are considered in the second paragraph of Sect. 5.1. It is stated now: "In the case of multi-layer clouds, individual layers could be hidden due to attenuation. Therefore only the uppermost cloud top and lowest base are considered."

**Comment:** - Referencing previous literature:
o The bi-modal distribution of trade cumuli in the vicinity of Barbados has been extensively studied by Nuijens et al. using data from the Barbados Cloud Observatory. Please refer to Nuijens et al. 2014 QJRMS in L56 and also later on in the manuscript. (Related to this, in L262 it would be good to mention that the 30% dominance of the upper mode vs. the lower mode is opposite when considering ground-based observations (see Nuijens et al. 2014).)

**Response:** We performed a more detailed literature review and included additional references. Related to Nuijens et al. (2014) we added after L56 from the initial submission: "In addition to spaceborne, also ground-based observations have been used to study the distribution of low-level cloud in the trades. Nuijens et al. (2014) analyzed cloud observations taken at the Barbados Cloud Observatory (BCO; Stevens et al., 2015) at the upstream eastern coast of Barbados at Deebles Point facing the Atlantic Ocean. For two years of ceilometer data of shallow clouds with tops below 4 km, they found that the shallow cloudiness is dominated by clouds near the lifting condensation level (LCL) with about two thirds of the shallow cloud coverage coming from clouds with bases below 1 km." Further, we added the following after L262 from the initial submission: "This is contrary to the ground-based impression from the same region but other period when the shallow mode was clearly dominating (Nuijens et al., 2014)."

**Comment:** o Observations from the CSET field campaign in the eastern pacific presented in O et al. 2018 GRL and Wood et al. 2018 JAS revealed the common occurrence of persistent thin outflow layers with very low droplet concentrations (the authors refer to 'veil' clouds and ultra-clean layers). It may be good to cite and discuss these papers in the context of the present results.

**Response:** Thanks for pointing this out. We follow these suggestions and could refer the observations of clouds in the outflow or stratocumulus mode with little LWP (below 10 and $50\,\mathrm{g\,m^{-2}}$) to such veil clouds. The second paragraph of Sect. 5.3 is extended to: "It is remarkable that high cloud tops in the stratiform layer were often observed by the lidar under low LWP conditions (below $10\,\mathrm{g\,m^{-2}}$). Such clouds likely correspond to thin 'veil' clouds frequently observed near the upper boundary layer, i.e., below the trade inversion, in the stratocumulus to cumulus transition by Wood et al. (2018) and O et al. (2018). They report on geometrically and optically thin clouds with low droplet number concentration (about $5\,\mathrm{cm^{-3}}$) but relatively large droplets with radii ranging from 15 to $30\,\mathrm{\mu m}$. Droplets of such sizes are large enough to provide a radar reflectivity above the detection threshold.

Extending the LWP class from $10\,\mathrm{g\,m^{-2}}$ to $50\,\mathrm{g\,m^{-2}}$ includes more additional lidar-detectable stratiform cloud coverage to the statistics than very shallow cloud coverage. This means even more veil clouds are included, which were estimated to have a typical LWP of about $25\,\mathrm{g\,m^{-2}}$ (Wood et al., 2018). In all cases with LWP $< 50\,\mathrm{g\,m^{-2}}$, the stratiform layer was observed about 1.5 times more often by the lidar than the layer of very shallow clouds, which is a bit more often than in the LEM and SRM (see also Fig. 8a)"

**Comment:** o The referencing for the first two sentences in the introduction should be improved.

**Response:** The references got mixed up. They should read: "The representation of low-level oceanic clouds contributes largely to differences between climate models in terms of equilibrium climate sensitivity (Bony and Dufresne, 2005; Schneider et al., 2017). Global atmospheric models with kilometer-scale resolution are considered the way forward in forecasting future climate scenarios (Satoh et al., 2019)."

**Comment:** - The LCL computation from the dropsondes: Can you say how many dropsondes are used to interpolate the LCL? And by how much they are separated in space and time on average?

**Response:** We added: "Fifty dropsondes were released in total in the study area with a median separation along flight track of about 515 km (quartiles: 384 and 658 km, see also Fig. 1)."

**Comment:** - Differences between the western and eastern part of the domain related to cloud deepening: Not only is there a difference in the height of the upper mode, but also in the normalized frequencies of the upper mode, with the deeper western half having a reduced frequency compared to the shallower eastern half. This, and also the insensitivity of the lower mode, was also shown in LES of Vogel et al. 2020 QJRMS.

**Response:** We appreciate this note. In the revised manuscript, we relate our findings to Vogel et al. (2020) and supplement our statement in the third paragraph of Sect. 5.2: "This deepening probably caused the frequency reduction of the stratiform mode in the western half compared to the shallower eastern half. Such relation between deepening of the cloud layer and reduced formation of stratiform clouds was also shown in an LES study by Vogel et al. (2020)." [...] "In contrast to the deeper and stratiform clouds, the frequency and height of very shallow lidar-visible clouds is almost the same in the western and eastern parts, which also agrees with Vogel et al. (2020)."

**Comment:** - Section 5.3: The discussion of the results in this section should be better structured and more focused on the most important features. It is not always clear what is compared to what, and there is a lot of switching around between LWP categories, the observations and the different models. I also spotted a lot of typographical errors that should be corrected (e.g. L347 partial coverage; L363 that such a cloud doesn't need any contribution...)

**Response:** Following also a comment by Referee #1, we realized that the discussion in Sect. 5.3 was not fully comprehensible. Thus, we reorganized section 35 and included an additional figure (Fig. 8), which shows the relation of key cloud features as a function of LWP more easily. This figure also shows the influence of model configuration (SRM or LEM) versus horizontal resolution by including the outputs of the SRM and LEM at two additional resolutions. Furthermore, We did a more thorough proof reading and revised typos.

**Comment:** - Figure 1: This figure could be improved. Please zoom more into the area of flight operations (only showing e.g. 7° N to 20° N), make sure that all flight paths are visible and not overlapping, and add markers/crosses for the dropsonde locations.

**Response:** We improved Fig. 1 following the suggestions. The partial overlap of flight tracks cannot be excluded completely, as the flights were accurately flown along the same track using the autopilot. However, we used different line widths to make sure that all tracks are visible.

r420

**Comment:** - Figure 5: What exactly does the cloud fraction in the legend refer to? Is it just the maximum cloud fraction? It would be nice to give the total projected cloud cover instead of a cloud fraction, as this would give a sense of the total cloudiness.

**Response:** The percentage in the parenthesis actually refers to the total projected 2D cloud coverage. To make this more clear,
r425 we added "2D" in several occasions and went through the manuscript to consistently use the terms "cloud coverage" for the projected 2D cloud coverage and "cloud fraction" for the vertically resolved profile of cloudiness.

**Comment:** - Figure A1: similar to the above, in the caption you mix cloud cover and cloud fraction, but I guess you mean the same thing.
r430 **Response:** Both should be the same. Thus, we changed "cloud cover" to "cloud fraction" in the caption.

**Comment:** TYPOGRAPHICAL SUGGESTIONS: - L65 (and everywhere else): model data –> model output
**Response:** We followed this suggestion.

r435 **Comment:** - L89: maybe add 'first phase of the NARVAL... Often referred to as NARVAL1 in other studies.
**Response:** We now mention NARVAL1 in parenthesis: "NARVAL-South (also referred to as NARVAL1)". However the term NARVAL1 is not used in the manuscript to not confuse with flights over the mid-latitude North Atlantic which where also part of the first NARVAL field experiment (Konow et al., 2019).

r440 **Comment:** - L97: this sentence is a bit odd there and should be moved down after L104, maybe adj. it to 'The following subsections describe...'
**Response:** We followed this suggestion.

**Comment:** - L98: form –> from
r445 **Response:** Corrected.

**Comment:** - L99: better than 20 g/m-2 and 10% (?)
**Response:** We now make clear, that this error is relative to the LWP itself, i.e. "The LWP retrieval from the microwave ra-diometer has a high accuracy, which is better than $20\,\mathrm{g\,m^{-2}}$ for LWP < $100\,\mathrm{g\,m^{-2}}$ and relatively better than 20 and $10\,\%$ of the
r450 retrieved LWP for LWP greater than $100\,\mathrm{g\,m^{-2}}$ and $500\,\mathrm{g\,m^{-2}}$, respectively, as described by Jacob et al. (2019)."

**Comment:** - L107: radar –> reflectivity
**Response:** We added "reflectivity" in "the radar reflectivity is approximately proportional".
.
r455 **Comment:** - L116: Ragged point –> Deebles point
**Response:** Corrected.

**Comment:** - L124: a clear frequency maximum?
**Response:** We clarified that the "frequency maximum" was meant.
r460

**Comment:** - L206 & other instances throughout the manuscript: remove commas before 'that' $\longrightarrow$ "it has to be noted that..."
**Response:** We removed the comma before "that" in several occasions.

**Comment:** - L212: by –> be

r465 **Response:** We corrected "can only by reached" to "can only be reached".

**Comment:** - L255-257: could be omitted.
**Response:** We omitted these last two sentences.

r470 **Comment:** - L285: what do you mean with shallow clouds here? (Please also see my specific comment on the definition of the cloud modes above)
**Response:** Now, we define cloud modes more clearly in the beginning of Sec. 5.

**Comment:** - L306: remove 'is'
r475 **Response:** Removed.

**Comment:** - L316: simulated infrequently –> underrepresented
**Response:** We changed "simulated infrequently compared to observations" to "underrepresented compared to observations".
**Comment:** - L321ff: The thresholds for the LWP classes are different from Figure 6 (i.e. lower bounds not given)
r480 **Response:** In the text, we included the lower threshold in the parenthesis to agree with that figure.

**Comment:** - L338: Ref to figure 5 and not figure 4
**Response:** Corrected.

r485 **Comment:** - L383: liar –> lidar
**Response:** Corrected.

**Comment:** - L419: a gap of what? Please be more specific
**Response:** It is the "gap in the cloud top frequency distribution".
r490
**Comment:** - L424: might be an...
**Response:** Corrected.

**Comment:** - L491: remove 'with'
r495 **Response:** Removed.

**Reply to Executive Editor Comment by Astrid Kerkweg**

**Comment:** Dear authors,
in my role as Executive editor of GMD, I would like to bring to your attention our Editorial version 1.2:
r500 https://www.geosci-model-dev.net/12/2215/2019/
This highlights some requirements of papers published in GMD, which is also available on the GMD website in the 'Manuscript Types' section:
http://www.geoscientific-model-development.net/submission/manuscript_types.html
In particular, please note that for your paper, the following requirements have not been met in the Discussions paper:

r505 If the model development relates to a single model then the model name and the version number must be included in the title of the paper. If the main intention of an article is to make a general (i.e. model independent) statement about the usefulness of a new development, but the usefulness is shown with the help of one specific model, the model name and version number must be stated in the title. The title could have a form such as, "Title outlining amazing generic advance: a case study with Model XXX (version Y)".

r510 As you are using just one model, please state in the title that you are using ICON and mention its version number.
Additionally, please provide more details on the used codes in the "Code availability" section: provide the numbers or unique identifiers of the code versions used and make sure that these versions (because this is relevant to reproduce the results of this article) are permanently archived and accessible in the future. With regard to the ICON model code I understand, that you did not perform the simulations yourself, but from your provided references (Klocke et al., 2017, and Vial et al., 2019) it is hard
r515 to find out (if at all) which model version(s) have been used and how to access the code. Please provide this information upon submission of the revised version.

**Response:** Two fundamentally different versions of ICON were used in this study. The ICON storm resolving model (SRM), which is typical run for numerical weather prediction, and the ICON-based Large Eddy Model (LEM), used by the research community on a case study bases. To make this clear, we would like to change the title to "Multi-layer Cloud Conditions in
r520 Trade Wind Shallow Cumulus – Confronting two ICON Model Derivates with Airborne Observations". Regarding the model versions, we specified that "The ICON SRM was run using revision '28436M' of the 'icon-nwp/icon-nwp-dev' branch (Klocke et al., 2017). The ICON LEM was run using the ICON release 2.3.00 (Stevens et al., 2019)." in the data availability section. If further details on the used model output are desired, please let us know, so we could make all ICON model output data we used available on a data archive.

r525 **Other minor changes**

We renamed the cloud cover variable $clc$ to "cloud fraction (CF)" in the appendix B.
We changed the abbreviation of the autoconversion to the more commonly used AU.
We renamed variable name for the droplet number concentration from $q_{nc}$ to the more commonly used "$N_c$".

[revised manuscript text omitted]

---

## Author Response (AR2)

**Response to „Topical Editor Decision: Publish subject to minor revisions (review by editor)"**

Dear Editor,

Thank you very much for reviewing our initial replies. Concerning your comment on the availability of the model code and output, we decided to upload the used model output to a long-term archive instead of referencing the code revisions. By this we ensure, that the results of the paper can be reproduced. As we did not perform the ICON model simulations ourselves, we believe that the archiving of the used output together with the referencing of the relevant articles is the best solution and we hope this meets your requirements.

Code and data availability before:

> The ICON SRM and LEM outputs were produced by Klocke et al. (2017) and made further public by Vial et al. (2019). The ICON SRM was run using revision "28436M" of the "icon-nwp/icon-nwp-dev" branch (Klocke et al., 2017). The ICON LEM was run using the ICON release 2.3.00 (Stevens et al., 2019).

Code and data availability now:

> The ICON SRM and LEM outputs were produced by Klocke et al. (2017) and made further public by Vial et al. (2019) and Stevens et al. (2019). The set of ICON model output used in this study is available at the long-term archive of the German Climate Computing Center (DKRZ; dataset DKRZ_LTA_834_ds00052 at https://cera-www.dkrz.de/WDCC/ui/cerasearch/entry?acronym=DKRZ_LTA_834_ds00052)

Furthermore, we removed the explanation of the acronym "DKRZ" from the acknowledgment section as it is now explained before in the code and data availability section.

[revised manuscript text omitted]